# Heterogeneous Quantum Federated Learning via Adaptive Circuit Search and Model Aggregation

## Abstract

Quantum federated learning (QFL) is an emerging framework for privacy-preserving, collaborative training of quantum neural networks across a network of quantum nodes operating under qubit resource constraints. Although promising, existing QFL approaches enforce a uniform quantum circuit architecture across nodes, failing to account for data heterogeneity and leading to suboptimal global model performance. To tackle these challenges, we propose a BO-QFL framework, which is based on Bayesian optimization to discover node-specific quantum circuit architectures and a novel aggregation rule to unify heterogeneous models at the quantum server. The novel contributions of this paper are twofold: (i) an adaptive circuit architecture search mechanism for heterogeneous quantum nodes, utilizing Bayesian optimization to automatically discover optimal quantum circuit configuration, and (ii) an effective and innovative aggregation strategy that integrates these locally optimized heterogeneous circuits into a unified global model through element-wise logical union. Through rigorous simulations on spatial and temporal datasets, the proposed framework demonstrates a significant improvement in the global model performance over fixed-architecture baselines. Additionally, evaluations in both noisy and ideal quantum environments further substantiate its robustness in realistic quantum settings.

## 1 Introduction

Quantum federated learning (QFL) is an advancing framework that enables distributed quantum devices (e.g., quantum processors) to collaboratively train a shared quantum model via local QNNs without sharing raw data Heidari et al. (2022) Huang et al. (2022). By decentralizing training, QFL addresses qubit-count limitations of current hardware, allowing each node (client) to train compact, device-compatible models that aggregate into a more expressive global model Araujo et al. (2024). Leveraging fundamental quantum properties and high-dimensional Hilbert spaces, QFL offers a promising path toward scalable, secure quantum machine learning with potential super-polynomial speedups Yamasaki et al. (2020).

A key determinant of QNN performance is the design of their parameterized quantum circuits (PQCs) Li et al. (2022); Barthe et al. (2025). Optimizing PQC architectures is challenging due to the vast space of gate choices, entanglement patterns, and hyperparameters, making automated search strategies essential. In classical ML, neural architecture search (NAS) methods, such as Bayesian optimization (BO) Cai et al. (2024); Fan et al. (2024), efficiently explore complex black-box spaces with limited evaluations. These ideas have recently extended to the quantum domain through quantum architecture search (QAS), enabling the discovery of effective PQC structures for quantum learning.

In QFL, existing search mechanisms typically assume a single uniform model across all quantum nodes. This assumption is often impractical under non-independent and identically distributed (non-IID) client data, where a circuit well-suited for one node may perform poorly for another, leading to inefficient resource use and degraded performance Gurung & Pokhrel (2025). This situation highlights *two primary research gaps: 1) The lack of methods for client-specific QNN architecture search in QFL, limiting adaptivity; 2) Even with such mechanism, heterogeneous model aggregation*

*remains a challenge, as conventional methods, such as FedAvg Collins et al. (2022), assume identical architectures, necessitating new aggregation schemes.* This leads to our primary research question: *How can we solve the dual challenges of client-specific architecture search and heterogeneous model aggregation within a unified QFL framework?* We address this by introducing BO-QFL, a novel unified framework that leverages BO to perform quantum node-specific architecture search and an innovative approach for heterogeneous model aggregation within the QFL system.

**Key Contributions:** The key contributions of this paper are summarized as follows.

- We design a Bayesian optimization-based QNN architecture search mechanism specifically for the QFL framework, which efficiently discovers a unique PQC architecture for each quantum node to accelerate the performance, considering a heterogeneous training environment. *To the best of our knowledge, we are the first to study BO-based adaptive circuit search within QFL.*
- We develop a novel adaptive aggregation strategy to address the resulting client model heterogeneity, which allows effective averaging from structurally diverse QNNs. *This is the first work to develop an effective aggregation strategy for heterogeneous quantum models within QFL.*
- Finally, with extensive simulations on real-world datasets, we assess the performance of BO-QFL, demonstrating its superiority over existing QFL systems.

## 2 RELATED WORKS

**QFL:** Research on QFL has advanced in a multitude of directions with a major focus on developing novel algorithms to enable efficient and scalable distributed training across a network of quantum devices Innan et al. (2024); Larasati et al. (2022). For instance, a recent study Zhang et al. (2025) addressed efficiency and security challenges in QFL by introducing a multi-qubit broadcast protocol and quantum state averaging. Similarly, Bhatia et al. (2024) addresses communication efficiency in QFL by leveraging quantum natural gradient descent within variational quantum circuits. Concurrently, a substantial number of studies have concentrated on implementing QFL in applications including healthcare Bhatia & Neira (2024), network systemsAraujo et al. (2024), and intrusion detection Yamany et al. (2021).

**Heterogeneity in QFL:** Heterogeneity in client data distributions and models has been shown to significantly degrade the QFL model performance, imposing a major bottleneck to real-world deployment Qu et al. (2022). The majority of QFL research Huang et al. (2022); Gurung et al. (2025); Subramanian & Chinnadurai (2024) assumes homogeneous quantum circuit architectures for all quantum nodes, which is unrealistic with varying task complexity, circuit depth, and hardware capabilities. While data heterogeneity from non-IID client distribution is widely researched in QFL, most studies simply analyze its effects without proposing true solutions. Authors in Zhao (2023) introduce a one-shot QFL framework using local density estimators, whereas Hisamori et al. (2024) rely on weighted model averaging, which merely limits the influence of poorly trained clients rather than improving performance.

**Quantum circuit search in QML/QFL:** Discovering an optimal circuit from a vast search space of gate arrangements and entanglement structures has become a pivotal challenge, driving significant research interest. For instance, authors in He et al. (2024); Sun et al. (2023) applied gradient-based optimization, while Dai et al. (2024) explored reinforcement learning. In parallel, Du et al. (2022); Zhang & Zhao (2023) assessed evolutionary algorithms, and stronger optimization techniques like genetic algorithms have also been explored Wei et al. (2021). Although promising, these algorithms cannot be directly applied to a QFL framework as a circuit optimized for one client's data or hardware is often suboptimal for others, intensifying model heterogeneity.

Despite such extensive research efforts, two critical gaps persist: *i) the existing literature largely overlooks the need for client-specific quantum neural network architecture searches within QFL and ii) even when diverse architectures are considered, traditional aggregation methods in the literature are unequipped to handle model heterogeneity.* Motivated by these gaps, we develop a novel QFL framework (BO-QFL) that uses BO to generate client-specific quantum circuits for improved local performance under data heterogeneity, along with a dedicated aggregation strategy to effectively integrate these heterogeneous models.

## 3 SYSTEM MODEL

We consider a QFL framework where $N$ heterogeneous quantum clients, denoted as $\mathcal{N} = \{1, 2, \ldots, N\}$, collaboratively search for optimal quantum circuit architectures. Subsequently, the distributed quantum devices/clients train a global quantum model, while the entire process is orchestrated by a central aggregator, as illustrated in the Fig. 1a. Each client $n \in \mathcal{N}$ holds a statistically diverse local dataset $D_n$, reflecting heterogeneous data distributions across the system. Each client performs local quantum architecture search over a predefined space of PQCs, represented by binary matrices where rows correspond to qubits and columns encode the presence of rotation gates $R_x$, $R_y$, and $R_z$ across the layers. Specifically, a PQC with $Q$ qubits and $L$ layers is encoded as $\mathbf{A}_n \in \{0, 1\}^{Q \times 3L}$, where each entry is defined as

$$\mathbf{A}_{n(q,g,l)} = \begin{cases} 1, & \text{if gate } g \in \{R_x, R_y, R_z\} \text{ is applied on qubit } q \text{ at layer } l, \\ 0, & \text{otherwise.} \end{cases} \quad (1)$$

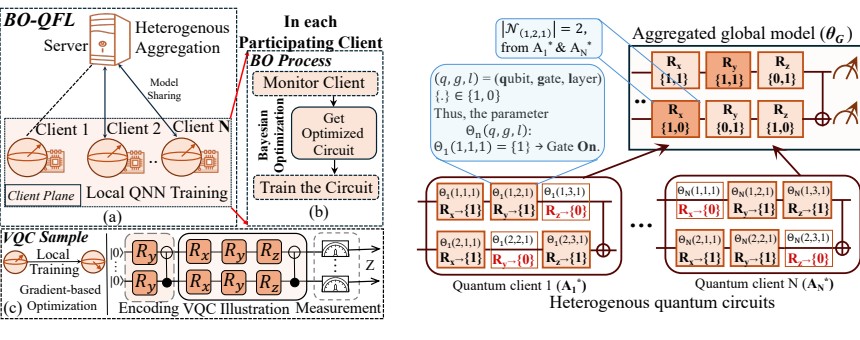

(a) System Overview        (b) Heterogenous Aggregation

Figure 1: Overall system architecture of the BO-QFL framework. (a) Overview of the BO-QFL system, where quantum clients perform BO-based local QNN training on non-IID data and send their optimized models to the server for heterogeneous aggregation into a global model. (b) Element-wise logical union for heterogeneous model aggregation, where each parameter in the global model is averaged exclusively over the subset of clients whose optimal architectures train that parameter.

Local architecture search at each client is carried out using BO, which leverages a Gaussian process (GP) surrogate to model the relationship between quantum circuit architectures and their corresponding accuracy. The objective for the client $n$ is to identify the optimal architecture by solving $\mathbf{A}_n^* = \arg\max_{\mathbf{A} \in \mathcal{S}} f_n(\mathbf{A}_n)$, where $f_n(\mathbf{A}_n) \in [0, 1]$ denotes the local accuracy and $S$ represents the predefined architecture search space. BO models the objective function with a GP prior as $f_n(\mathbf{A}_n) \sim \mathcal{GP}\big(m(\mathbf{A}_n), k(\mathbf{A}_n, \mathbf{A}_n')\big)$, where $m(\cdot)$ and $k(\cdot, \cdot)$ denote the mean and kernel functions, respectively. Here, $\mathbf{A}_n$ represents the current architecture under evaluation, and $\mathbf{A}_n'$ denotes another architecture in the search space, utilized for computing correlations through the kernel function $k(\cdot, \cdot)$. This GP prior provides a probabilistic model for the objective function, which is then used to construct an acquisition function for sequential sampling. At each BO iteration, the next architecture is selected by maximizing the logarithm of the expected improvement (EI) as $\mathbf{A}_{n,\text{next}} = \arg\max_{\mathbf{A} \in \mathcal{S}} \log \text{EI}(\mathbf{A}_n)$. We mention that the GP surrogate predicts the function distribution, while the acquisition function (second equation) determines the most promising architecture for evaluation based on this prediction. This process continues until a stopping criterion is met, such as reaching a maximum number of iterations, achieving a predefined accuracy threshold, or observing convergence of the acquisition function. After all clients complete BO, the server constructs a global architecture $\mathbf{A}_G$ by performing a union of local architectures as $\mathbf{A}_G(q, g, l) = 1$ if $\exists n \in \mathcal{N}$ such that $\mathbf{A}_n^*(q, g, l) = 1$, where $(q, g, l)$ denotes the qubit, gate type, and layer indices, respectively. Equivalently, the global architecture can be expressed as an element-wise logical OR as $\mathbf{A}_G = \bigvee_{n=1}^{N} \mathbf{A}_n^*$. The global PQC is then initialized and distributed to all clients. For weight initialization, each client adopts the global weights for gates common to its local and global architectures, while the remaining gates are randomly initialized. Particularly,

$$\Theta_n(q, g, l) = \begin{cases} \Theta_G(q, g, l), & \text{if } (q, g, l) \in \mathbf{A}_n^* \cap \mathbf{A}_G, \\ \text{random init}, & \text{otherwise,} \end{cases} \quad (2)$$

where $\Theta_n(q, g, l)$ is the trainable parameter of client $n$ for the gate at qubit $q$, gate type $g$, and layer $l$. Each client trains its PQC locally for $E$ epochs, where the gradients are estimated using the parameter-shift rule Wang et al. (2022) and optimized using the Adam optimizer Zhang et al. (2022). Let $\theta_n^t$ denote the client's model parameters at iteration $t$, $\eta$ the learning rate, and $\mathcal{L}$ the local loss function. The update rule is expressed as $\theta_n^{t+1} = \theta_n^t - \eta\nabla\mathcal{L}(\theta_n^t)$. After local training, clients return updated parameters corresponding to gates shared with the global architecture. The server aggregates these weights using an averaging rule $\Theta_G(q, g, l) = \frac{1}{|\mathcal{N}_{(q,g,l)}|}\sum_{n\in\mathcal{N}_{(q,g,l)}}\Theta_n(q, g, l)$, where $\mathcal{N}_{(q,g,l)} = \{n : (q, g, l) \in \mathbf{A}_n^*\}$, denotes the set of clients that include the gate at position $(q, g, l)$ in their local architecture. Subsequently, these aggregated parameters are unified to generate a single global model $\theta_G$ and redistributed to the quantum clients for further training. This process of aggregation and redistribution is repeated until convergence.

## 4 METHODOLOGY

### 4.1 PROPOSED BO METHOD

To efficiently navigate the vast and computationally expensive search space of PQC architectures, we employ a sample-efficient optimization strategy. We adopt a Bayesian Optimization (BO) framework where the objective function $f(\mathbf{x})$ maps a circuit architecture $\mathbf{x}$ to its test accuracy. The search begins with an initial dataset $\mathcal{D}_0$ of $n_0$ architectures sampled from a continuous domain $[0, 1]^{Q\times 3L}$ using a Sobol sequence and rounded to a binary format. To model this expensive-to-evaluate function, we use a Gaussian Process (GP) surrogate, $f(\mathbf{x}) \sim \mathcal{GP}(m(\mathbf{x}), k(\mathbf{x}, \mathbf{x}'))$, with a scaled Radial Basis Function (RBF) kernel given by $k(\mathbf{x}, \mathbf{x}') = \sigma_f^2\exp\left(-\frac{1}{2\ell^2}\|\mathbf{x} - \mathbf{x}'\|^2\right)$. The GP hyperparameters are optimized by maximizing the marginal log-likelihood: $\log p(\mathbf{f} \mid \mathbf{X}) = -\frac{1}{2}\mathbf{f}^\top(\mathbf{K} + \sigma_n^2\mathbf{I})^{-1}\mathbf{f} - \frac{1}{2}\log\det(\mathbf{K} + \sigma_n^2\mathbf{I}) - \frac{n_0}{2}\log(2\pi)$. The fitted GP provides a posterior predictive distribution $f(\mathbf{x}_*) \mid \mathcal{D}_0 \sim \mathcal{N}(\mu(\mathbf{x}_*), \sigma^2(\mathbf{x}_*))$, with mean $\mu(\mathbf{x}_*) = \mathbf{k}_*^\top(\mathbf{K} + \sigma_n^2\mathbf{I})^{-1}\mathbf{f}$ and variance $\sigma^2(\mathbf{x}_*) = k(\mathbf{x}_*, \mathbf{x}_*) - \mathbf{k}_*^\top(\mathbf{K} + \sigma_n^2\mathbf{I})^{-1}\mathbf{k}_*$. This posterior guides the search by selecting the next candidate that maximizes the Log Expected Improvement (LogEI) acquisition function, $\mathrm{LogEI}(\mathbf{x}) = \log(\mathrm{EI}(\mathbf{x})+1)$, to balance exploration and exploitation. This iterative process of updating the surrogate and selecting new candidates continues until a stopping criterion is met, ensuring a thorough yet bounded search.

### 4.2 HETEROGENEOUS MODEL AGGREGATION

To address the challenge of aggregating parameters from structurally diverse quantum circuits, we introduce a strategy where the global model's architecture is defined using an element-wise logical union of all client architectures. To aggregate corresponding parameters, the global model is decomposed into components, where each component corresponds to the group of clients that trained it (Fig. 1b). This ensures each component is updated only by those clients, stabilizing the process and preserving specialized knowledge. Let the universe of all gate positions be $\mathcal{P} = \{(q, g, l) \mid 1 \le q \le Q, g \in \{R_x, R_y, R_z\}, 1 \le l \le L\}$. The active parameter set for client $n$, denoted $\mathcal{W}_n^*$, is $\mathcal{W}_n^* = \{p \in \mathcal{P} \mid \mathbf{A}_n^*(p) = 1\}$, and the set of all parameters in the global model is $\mathcal{W}_G = \bigcup_{n=1}^N \mathcal{W}_n^*$. For each $p \in \mathcal{W}_G$, the set of clients containing $p$ is $\mathcal{N}_p = \{n \in \mathcal{N} \mid p \in \mathcal{W}_n^*\}$. The global parameter update is $\Theta_G^{(k+1)}(p) = \frac{1}{|\mathcal{N}_p|}\sum_{n\in\mathcal{N}_p}\Theta_n^{(k)}(p)$. Equivalently, expanding $p = (q, g, l)$ gives

$$\Theta_G^{(k+1)}(q, g, l) = \frac{\sum_{n=1}^N \Theta_n^{(k)}(q, g, l) \cdot \mathbf{A}_n^*(q, g, l)}{\sum_{n=1}^N \mathbf{A}_n^*(q, g, l)}. \tag{3}$$

The final global model is reconstructed by performing this operation for every $p \in \mathcal{W}_G$, forming the unified $\theta_G^{(k+1)}$.

Algorithm 1 summarizes BO-QFL. Clients first perform BO with LogEI to obtain their optimal architectures $\mathbf{A}_n^*$ (lines 3–5). The server then forms the global union $\mathbf{A}_G$ and initializes parameters as in Eq. equation 2 (lines 7–8). In each round (lines 10–15), clients train locally for $T$ epochs, send updates, and the server aggregates and redistributes parameters. This repeats for up to $K$ global rounds until convergence. A detailed version of the algorithm is given in Appendix A.11.

---

**Algorithm 1** BO QFL with client-specific architecture search and heterogeneous aggregation

---

1: **Input:** clients $\mathcal{N} = \{1, \ldots, N\}$, datasets $\{D_n\}$, search space $\mathcal{S} \subset \{0, 1\}^{Q \times 3L}$, BO budget $E$ with seeds $n_0$, total global rounds $K$, total local epochs per round $T$
2: **Output:** global union architecture $\mathbf{A}_G$, global parameters $\Theta_G^{(K)}$
3: **Client side BO search, parallel over** $n \in \mathcal{N}$
   Sample $n_0$ seeds, train and score on $D_n$, then iterate $e = n_0 + 1, \ldots, E$: fit GP, select next by LogEI, train and score
   Set $\mathbf{A}_n^*$ to the best scoring architecture
4: **Build union architecture and initialize**
   Form $\mathbf{A}_G$ via union rule
   Initialize client parameters using the case rule, see Eq. equation 2
5: **Federated training with heterogeneous aggregation**
6: **for** $k = 0$ to $K - 1$ **do**
7:    **Local update, parallel over** $n$: train for $t = 1, \ldots, T$ epochs on $D_n$ starting from $\Theta_n^{(k)}$, produce $\Theta_n^{(k+1)}$
8:    **Server aggregation**: update $\Theta_G^{(k+1)}$ using the heterogeneous averaging rule, see Eq. equation 3
9:    **Broadcast**: send $\Theta_G^{(k+1)}$ to all clients and align supports with $\mathbf{A}_n^*$
10:   **if** convergence criterion holds **then break**
11:   **end if**
12: **end for**
13: **return** $\mathbf{A}_G, \Theta_G^{(k+1)}$

---

# 5 CONVERGENCE ANALYSIS

We conduct a rigorous convergence analysis of BO-QFL framework under full-device participation, accounting for non-convex loss functions, heterogeneous data distributions, and quantum shot noise.

## 5.1 COMPLEXITY ANALYSIS

Under the assumption of $L$-smoothness and $\mu$-PL Ajalloeian & Stich (2020), for $\boldsymbol{\theta}^0$ and $\eta_k = \mu \leq \frac{1}{L}$: $\mathbb{E}[f_n(\boldsymbol{\theta}^T)] - L^* \leq (1 - \eta\mu)^T ([f_n(\boldsymbol{\theta}^0)] - L^*) + \frac{1}{2}[\frac{\eta L V}{\mu}]$, where $V = \frac{\nu N_z D Tr(Z^2)}{2H}$. Given some target error level $\delta > 0$, for learning rate $\eta = \eta^{\text{shot-noise}} \leq \min\{\frac{1}{L}, \frac{\delta\mu}{LV}\}$, a number of iteration, given as $T^{\text{shot-noise}} = \mathcal{O}(\log\frac{1}{\delta} + \frac{V}{\delta\mu})\frac{L}{\mu}$, is sufficient to ensure an error $\mathbb{E}[f_n(\boldsymbol{\theta}^T) - f_n^*] = \mathcal{O}(\delta)$.

## 5.2 NOTATION AND DEFINITION

We study the problem $\min_{\boldsymbol{\theta}} f(\boldsymbol{\theta}) \triangleq \sum_{n=1}^N f_n(\boldsymbol{\theta})$, where $f(\boldsymbol{\theta})$, where each client $n$ trains on dataset $\mathcal{S}_n$ of size $S_n$ sampled from $\mathcal{D}_n$. The full gradient is $g_n \overset{\triangle}{=} \frac{1}{|\mathcal{S}_n|}\nabla f(\boldsymbol{\theta}; \mathcal{S}_n)$, and the stochastic gradient as $\tilde{g}_n \overset{\triangle}{=} \frac{1}{B}\nabla f(\boldsymbol{\theta}; \xi_n)$, with $\xi_n \subseteq \mathcal{S}_n$, $|\xi_n| = B$. Let $g_{n,k}^t$ and $\tilde{g}_{n,k}^t$ denote the full and stochastic gradients at round $t$. Each local parameter $\boldsymbol{\theta}_{n,k}^t$ is reparameterized into a union architecture $A_G$, with global average $\bar{\boldsymbol{\theta}}_k^t \overset{\triangle}{=} \frac{1}{N}\sum_{n\in\mathcal{N}} \boldsymbol{\theta}_{n,k}^{t,G}$, where $\boldsymbol{\theta}_{n,k}^{t,G}$ denotes the reparameterized form of client $n$'s parameters projected into the global architecture $A_G$. Moreover, $\tilde{g}_k^t \overset{\triangle}{=} \frac{1}{N}\sum_{n\in\mathcal{N}} \tilde{g}_{n,k}^t$, $g_k^t \overset{\triangle}{=} \frac{1}{N}\sum_{n\in\mathcal{N}} g_{n,k}^t$. Thus, the local SGD update at device $n$ is followed as $\boldsymbol{\theta}_{n,k}^{t+1} = \boldsymbol{\theta}_{n,k}^t - \eta_k \tilde{g}_{n,k}^t$, while $\bar{\boldsymbol{\theta}}_k^{t+1} = \bar{\boldsymbol{\theta}}_k^t - \eta_k \tilde{g}_k^t$ and $\mathbb{E}\tilde{g}_k^t = g_k^t$, where $\mathbb{E}$ represents function's expectation. We assume that $\frac{\sum_{n=1}^N ||g_{n,k}^t||_2^2}{||\sum_{n=1}^N g_{n,k}^t||_2^2} \leq \lambda$. The architecture divergence is $\psi_k^t = \frac{1}{N}\sum_{n=1}^N ||\boldsymbol{\theta}_{n,k}^{t,G} - \bar{\boldsymbol{\theta}}_k^t||^2$. The BO suboptimality gap at round $t$ is $\epsilon_{\text{BO}}^t \triangleq \mathbb{E}[\gamma(\bar{\boldsymbol{\theta}}_k^t, a^{t,*}) - \gamma(\bar{\boldsymbol{\theta}}_k^t, a^t)]$, where $a^t$ is the architecture chosen by BO, and $a^{t,*}$ is the optimal architecture. The BO regret over $T$ rounds is $R_T = \sum_{t=0}^{T-1} \epsilon_{\text{BO}}^t$. We now outline the key assumptions that form the basis of our convergence analysis.

**Assumption 1** (Smoothness and Lower Boundedness). *The $f_n(.)$ associated with device $n$ is differentiable for $1 \leq n \leq N$ and is $L - smooth$, i.e., $||\nabla f_n(\mathbf{u}) - \nabla f_n(\mathbf{v})|| \leq L||\mathbf{u} - \mathbf{v}||, \forall \mathbf{u}, \mathbf{v} \in \mathbb{R}^d$.*

**Assumption 2** ($\mu$-Polyak-Lojasiewicz (PL)). *The global objective function $f(.)$ is differentiable and satisfy the Polyak-Lojasiewicz (PL) condition with constant $\mu$, i.e., $\frac{1}{2}||\nabla f(\boldsymbol{\theta})||_2^2 \geq \mu(f(\boldsymbol{\theta}) - f(\boldsymbol{\theta}^*))$ holds $\forall \boldsymbol{\theta} \in \mathbb{R}^d$ with $\boldsymbol{\theta}^*$ being the optimal solution of global objective.*

**Assumption 3** (Bounded Local Variance). *For every local dataset $S_n$, $n = 1, 2, \ldots, N$, we can sample $\xi_n \subseteq S_n$ with $|\xi_n| = B$ and compute $\tilde{g}_n = \frac{1}{B}\nabla f(\boldsymbol{\theta}; \xi_n)$, $\mathbb{E}[\tilde{g}_n] = g_n = \frac{1}{|S_n|}\nabla f(\boldsymbol{\theta}; S_n)$ with the variance bounded as $\mathbb{E}[||\tilde{g}_n - g_n||^2] \leq C_1||g_n||^2 + \frac{\sigma^2}{B}$, where $C_1$ and $\sigma$ are constants.*

**Assumption 4** (BO Suboptimality Bias). *The estimator $\tilde{g}_{n,k}^t$ may deviate such that $\mathbb{E}[\tilde{g}_{n,k}^t] = \nabla f_n(\boldsymbol{\theta}_{n,k}^t) + e_{n,k}^t$, where $||e_{n,k}^t|| \leq \epsilon_{BO}$ bounds the deviation induced by the BO selection.*

**Assumption 5** (Architecture Divergence). *Let the local model $\boldsymbol{\theta}_{n,k}^t$ for client $n$ differ from the global mean $\bar{\boldsymbol{\theta}}_k^t$, then $\psi_k^t := \sum_{n=1}^N ||\boldsymbol{\theta}_{n,k}^t - \bar{\boldsymbol{\theta}}_k^t||^2$ and $||\nabla f_n(\boldsymbol{\theta}_{n,k}^t) - \nabla f_n(\bar{\boldsymbol{\theta}}_k^t)|| \leq L||\boldsymbol{\theta}_{n,k}^t - \bar{\boldsymbol{\theta}}_k^t||$.*

From the update rule and assumption on the L-smoothness of the objective function, we have $f(\bar{\boldsymbol{\theta}}_k^{t+1}) - f(\bar{\boldsymbol{\theta}}_k^t) \leq -\eta_k\langle\nabla f(\bar{\boldsymbol{\theta}}_k^t), \tilde{g}_k^t\rangle + \frac{\eta_k^2 L}{2}||\tilde{g}_k^t||^2$. Now, we take expectation on both sides of the inequality results in $\mathbb{E}[f(\bar{\boldsymbol{\theta}}_k^{t+1}) - f(\bar{\boldsymbol{\theta}}_k^t)] \leq -\eta_k\mathbb{E}[\langle\nabla f(\bar{\boldsymbol{\theta}}_k^t), \tilde{g}_k^t\rangle] + \frac{\eta_k^2 L}{2}\mathbb{E}[||\tilde{g}_k^t||^2]$ By taking the average for all the local and global iterations, we get

$$\frac{1}{KT}\sum_{k=1}^K\sum_{t=1}^T\mathbb{E}[f(\bar{\boldsymbol{\theta}}_k^{t+1}) - f(\bar{\boldsymbol{\theta}}_k^t)] \leq \frac{1}{KT}\sum_{k=1}^K\sum_{t=1}^T(-\eta_k\mathbb{E}[\langle\nabla f(\bar{\boldsymbol{\theta}}_k^t), \tilde{g}_k^t\rangle]) + \frac{1}{KT}\sum_{k=1}^K\sum_{t=1}^T\frac{\eta_k^2 L}{2}\mathbb{E}[||\tilde{g}_k^t||^2]. \tag{4}$$

Next, we bound each term in equation 4: Lemma 1 handles the first term via gradient–stochastic alignment, Lemma 3 bounds the second, and Lemma 2 addresses the residual term from Lemma 1. We begin by presenting key lemmas that form the basis of our main result.

**Lemma 1.** *Let Assumption 1 hold, in the BO-QFL framework, the expected inner product between stochastic gradient and full gradient is bounded by $-\eta_k\mathbb{E}(\langle\nabla f(\bar{\boldsymbol{\theta}}_k^t), \tilde{g}_k^t\rangle) \leq -\frac{\eta_k}{2}||\nabla f(\bar{\boldsymbol{\theta}}_k^t)||^2 - \frac{\eta_k}{2}||\sum_{n=1}^N\nabla f_n(\boldsymbol{\theta}_{n,k}^t)||^2 + \sum_{n=1}^N L^2||\bar{\boldsymbol{\theta}}_k^t - \boldsymbol{\theta}_{n,k}^t||_2^2 + \eta_k\epsilon_{BO} + \frac{\eta_k L}{2}\Psi_k^t$.*

*Proof.* See A.1 in the Appendix.

**Lemma 2.** *Let Assumption 3 hold, the expected upper bound of the divergence of $\boldsymbol{\theta}_{n,k}^t$ is given as $\frac{1}{KT}\sum_{k=1}^K\sum_{t=1}^T\sum_{n=1}^N\left[\mathbb{E}||\bar{\boldsymbol{\theta}}_k^t - \boldsymbol{\theta}_{n,k}^t||\right] \leq \frac{(2C_1+T(T+1))}{KT}\eta_k^2\frac{N+1}{N}\frac{1}{KT}\sum_{k=1}^K\sum_{t=1}^T\sum_{n=1}^N||g_{n,k}^t||^2 + \frac{\eta_k^2(N+1)(T+1)\sigma^2}{NB} \leq \frac{\lambda\eta_k^2(2C_1+T(T+1))}{KT}\frac{N+1}{N}\frac{1}{KT}\sum_{k=1}^K\sum_{t=1}^T\sum_{n=1}^N||g_{n,k}^t||^2 + \frac{\eta_k^2 KT(N+1)(T+1)\sigma^2}{NB}$.*

*Proof.* See A.2 in the Appendix.

**Lemma 3.** *Under Assumption 3, the expected upper bound of $\mathbb{E}[||\tilde{g}_k^t||^2]$ is expressed as $\mathbb{E}\left[||\tilde{g}_k^t||^2\right] \leq \lambda\left(\frac{C_1}{N} + 1\right)\left[\sum_{n=1}^N||\nabla f_n(\boldsymbol{\theta}_{n,k}^t)||^2\right] + \frac{\sigma^2}{NB} + L^2\Psi_k^t$.*

*Proof.* See A.3 in the Appendix.

**Lemma 4.** *For variance of the gradient estimate, $var(\xi_k^t) \leq \frac{1}{N}\sum_{n\in\mathcal{N}}\frac{\nu N_z DTr(Z^2)}{2H}$.*

*Proof.* See A.4 in the Appendix.

**Theorem 1.** *Let Assumptions 1, 2, 3 hold, then the upper bound of the convergence rate of the global model training considering full device participation after $K$ global rounds satisfies $\frac{1}{KT}\sum_{k=1}^K\sum_{t=1}^T\mathbb{E}||\nabla f(\bar{\boldsymbol{\theta}}_k^t)||^2 \leq \frac{2[f(\bar{\boldsymbol{\theta}}_1^0) - f^*]}{\eta_k KT} + \frac{L\eta_k\sigma^2}{NB} + \frac{2\eta_k^2\sigma^2 L^2(T+1)}{B}\left(1 + \frac{1}{N}\right) + \frac{1}{N}\sum_{n\in\mathcal{N}}\frac{\nu N_z DTr(Z^2)}{2H} + \frac{2}{KT}\sum_{k,t}\epsilon_{BO}^t + \frac{L}{KT}\sum_{k,t}\Psi_k^t$.*

*Proof.* See A.5 in the Appendix.

**Remark 1.** *The convergence bound in Theorem 1, for full device participation, shows dependence on the number of global rounds, participating clients $N$, and quantum measurement shots per client.*

**Remark 2.** *Theorem 1 indicates that increasing $N$ improves gradient averaging, while higher shot counts reduce variance. The bound also reflects noise accumulation across clients, so aggressive scaling in $N$ or shots yields diminishing returns. Balanced increases in both are more effective.*

**Remark 3.** *In BO-QFL, convergence is further affected by* architecture divergence*, mismatches among heterogeneous client models,* BO suboptimality*, deviations between selected and ideal architectures. These can slow convergence unless mitigated by careful global PQC design and robust BO strategies.*

## 6 SIMULATIONS AND RESULTS

### 6.1 DATASETS AND EXPERIMENTAL SETTING

Three datasets are chosen to cover different data types, two for image-based tasks and another for time-series, to stress the framework across both spatial and sequential inputs.

**MNIST:** The MNIST dataset Deng (2012) consists of 70,000 grayscale images of handwritten digits, 28 by 28 pixels, 60,000 training images, and 10,000 testing images. Each data point was flattened to a vector of length 784 and subjected to a deep-layer, resulting in a real-valued vector of length 1024 for amplitude encoding with 10 qubits ($2^{10} = 1024$). **HAR:** The Human Activity Recognition (HAR) dataset Reyes-Ortiz et al. (2013) that we used is from the UC Irvine Machine Learning Repository. Both time and frequency domain data were collected via 30 volunteers engaging in 6 activities, making up the classes: Walking, Walking Upstairs, Walking Downstairs, Sitting, Standing, and Lying. To prepare this vector for quantum processing, we reduced the feature vector size from a length of 561 to 256 (making $2^8$) and employed amplitude encoding with 8 qubits to map it onto the quantum state. **Fashion MNIST:** To put real stress on the designed framework, we use the Fashion-MNIST dataset Xiao et al. (2017), a more complex replacement for MNIST. It contains 70,000 28x28 grayscale images across 10 clothing categories. The preprocessing, distribution, and encoding procedures are analogous to those of MNIST.

**Key hyperparameters:** *PQC architecture*: Qubits: 8 for HAR, 10 for MNIST and Fashion MNIST; Layers: 4, each layer rotational gates then CNOT; Encoding: amplitude; Measurement: Pauli-Z; Gates searched: $\{R_X, R_Y, R_Z\}$; Entanglement: ring CNOT; Noise: shot noise from $H$ measurements, depolarizing probability 0.03 to 0.05%.
*BO search per client*: Iterations: 30 with early stopping; Initial points: 3 Sobol; Objective: local test accuracy; Acquisition Log Expected Improvement.
*QFL training*: Global rounds: 50; Local epochs: 5; Batch size: 64.
*Optimizer all stages*: Optimizer: Adam; Learning rate $5 \times 10^{-3}$; Loss: Negative Log Likelihood.

All experiments presented in this paper were conducted on a single GPU system equipped with an NVIDIA GeForce RTX 4090 GPU, 64 GB of RAM, and running Ubuntu 22.04. The simulation environment was built in Python, utilizing PyTorch for deep learning structures, TorchQuantum for quantum circuit simulation, and the BoTorch and GPyTorch libraries for implementing the Bayesian optimization. In our simulations, we use a network of 3, 6, or 12 quantum clients, each with a non-IID data distribution to reflect realistic decentralized learning environments.

### 6.2 EVALUATION ON A SINGLE CLIENT

Before evaluating our full BO-QFL system, we first validate the performance of the BO-based architecture search on a single client. To this end, we first benchmark it against a leading state-of-the-art RL-based approach for quantum architecture search. A Deep Q-Network (DQN) agent with an epsilon-greedy (from 1 to 0.05) exploration strategy and a 10,000-sample experience replay buffer is utilized for a robust RL system. The key difference is that the DQN agent sequentially builds a circuit by placing individual gates, whereas our BO method holistically evaluates entire architectures at once using a global probabilistic model of the search space. We also compare these frameworks with a baseline QNN where each layer consists of an $R_y$ gate and a CNOT gate per qubit wire.

Fig. 2 shows the test accuracies for architectures trained on a single client, comparing results for MNIST (a), HAR (b), and Fashion-MNIST (c). Both the BO and RL-optimized models surpass

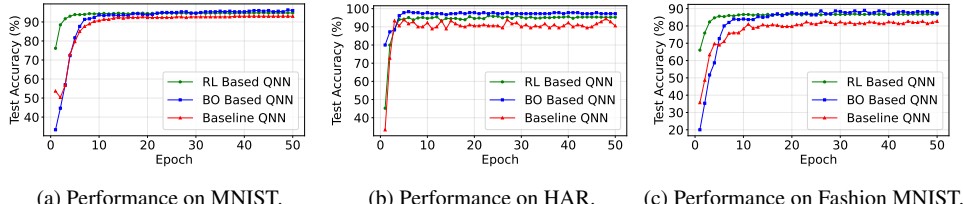

(a) Performance on MNIST.  (b) Performance on HAR.  (c) Performance on Fashion MNIST.

Figure 2: Performance comparison for QNNs optimized using RL, BO, and baseline architectures. Results show the test accuracy evolution across epochs for (a) MNIST, (b) HAR, and (c) Fashion MNIST.

the baseline QNN by a significant 2-8%. The performance of these optimized models is largely on par, with BO slightly outperforming RL by a small margin of ∼2% on the more complex HAR and Fashion-MNIST datasets. This demonstrates that both methods successfully find high-performing quantum circuits tailored to client-specific data dynamics.

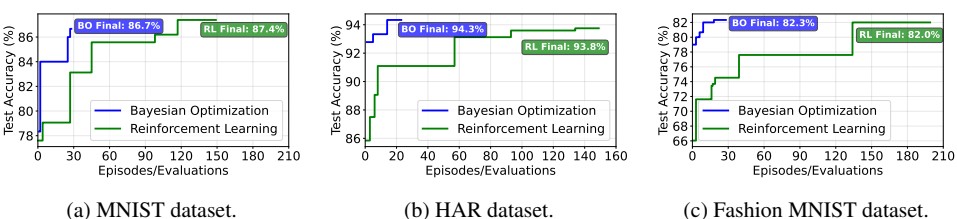

(a) MNIST dataset.  (b) HAR dataset.  (c) Fashion MNIST dataset.

Figure 3: Comparison of running-best test accuracy for RL-based and BO-based quantum architecture search on (a) MNIST, (b) HAR, and (c) Fashion MNIST. The plots show the cumulative best test accuracy achieved as a function of the number of optimization episodes, demonstrating the convergence speed and final performance of each optimization method.

However, the superiority of BO is evident in its search efficiency, as demonstrated in Fig. 3. Although their final architectures are comparable in model performance, the RL-based search requires dramatically more evaluations to converge. Across all datasets, the peak-performing RL architectures were found after at least 120 episodes, whereas BO was able to identify its optimal circuits within just 30 rounds. While each evaluation in both methods involved training a candidate circuit for 30 local epochs, the fundamentals of the BO search remain computationally lightweight. In contrast, each RL episode requires more complex agent-environment interactions, making the efficiency of BO a key advantage.

### 6.3 BO-QFL PERFORMANCE EVALUATION

We now evaluate the full BO-QFL framework within a 3-client system against two baselines under the non-IID data distributions. As seen in the Fig. 4, the first is QFL (No Aggregation), where FedAvg fails under model heterogeneity by excluding clients with mismatched PQC structures, resulting in unstable training and poor convergence. The second is the Baseline QFL, a homogeneous setup with identical non-optimized $R_y$ gate-based circuits across all clients. BO-QFL method consistently surpasses the standard baseline, improving global accuracy by 5–6% on MNIST (Fig. 3a), 11–12% on HAR (Fig. 3b), and 5% on Fashion-MNIST (Fig. 3c), while avoiding the learning failure seen in the "no-aggregation" case. This confirms that the BO-based method successfully finds tailored client circuits and our novel aggregation effectively unifies them.

Table 1 empirically validates our theoretical remarks. As in Remark 2, final accuracy depends on both the number of clients ($N$) and measurement shots. In the ideal case, increasing $N$ improves accuracy (e.g., MNIST BO-QFL: 91.78% to 94.61%) due to better gradient averaging, but in noisy settings this trend reverses (86.50% to 81.92%), confirming the impact of accumulated quantum noise. Higher shot counts consistently help, as 150-shot results surpass 100-shot results. Consistent with Remark 3, BO-QFL significantly outperforms Baseline QFL, showing the superiority of optimized architectures despite the BO's suboptimality in some cases. The noisy accuracy drop with

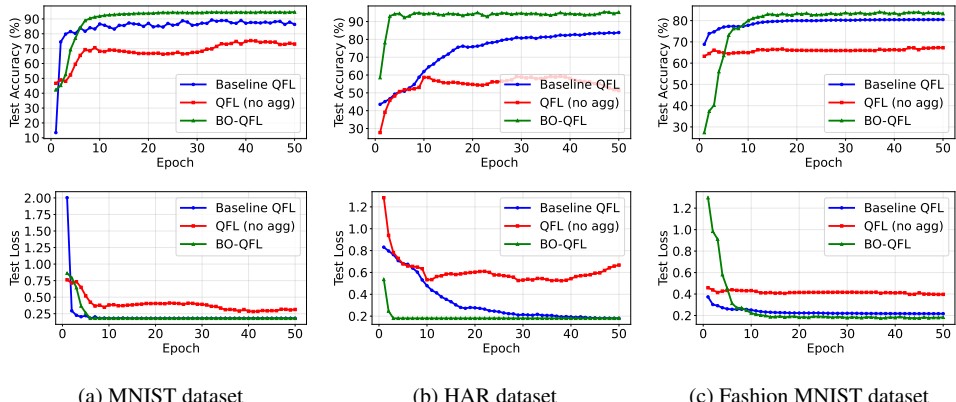

(a) MNIST dataset      (b) HAR dataset      (c) Fashion MNIST dataset

Figure 4: Performance comparison of BO-QFL approaches, where each subplot group shows the test accuracy (top) and loss (bottom) evolution of Baseline QFL, QFL (no agg), and BO-QFL methods on (a) MNIST, (b) HAR, and (c) Fashion-MNIST datasets, demonstrating the effectiveness of BO in quantum federated settings.

Table 1: Performance and scalability analysis of BO-QFL and Baseline QFL with different datasets and shot counts in ideal and noisy quantum simulation environments.

| Dataset | # Clients | BO-QFL Accuracy (%) | | | QFL (Baseline) Accuracy (%) | | |
|---|---|---|---|---|---|---|---|
| | | Ideal | Noisy (Shots=150) | Noisy (Shots=100) | Ideal | Noisy (Shots=150) | Noisy (Shots=100) |
| MNIST | 3 | 91.78 | **86.50** | **82.17** | 86.32 | **80.75** | 75.93 |
| | 6 | 92.61 | 84.39 | 80.45 | 84.18 | 79.11 | **76.21** |
| | 12 | **94.61** | 81.92 | 77.58 | **88.07** | 76.84 | 72.50 |
| HAR | 3 | 95.19 | **88.23** | 84.61 | 83.83 | **77.48** | **73.15** |
| | 6 | 93.80 | 86.91 | 83.05 | **89.42** | 75.99 | 71.82 |
| | 12 | **95.21** | 85.05 | 81.19 | 81.90 | 74.13 | 72.76 |
| Fashion-MNIST | 3 | 84.44 | 77.16 | **74.88** | 79.33 | 73.51 | **69.95** |
| | 6 | 83.28 | **77.72** | 73.50 | **84.50** | **73.88** | 68.74 |
| | 12 | **85.42** | 75.69 | 71.93 | 80.71 | 70.94 | 66.52 |

larger $N$ further underscores the architectural divergence, where aggregating more diverse, noisy models degrades the overall global performance.

# 7 CONCLUSION

In this work, we introduced the BO-QFL framework to overcome the limitations of homogeneous models in non-IID QFL by integrating a Bayesian Optimization search for client-specific quantum circuits with a novel strategy for heterogeneous aggregation. Simulations show our framework significantly outperforms standard baselines in both ideal and noisy settings and is significantly more sample-efficient than a state-of-the-art reinforcement learning approach. This work demonstrates a practical path toward adaptive and efficient QFL systems capable of handling the architectural and data heterogeneity of real-world decentralized quantum networks. Future work could focus on extending this framework to include hardware-aware optimizations, further bridging the gap to deployment on real-world quantum hardware.

## REPRODUCIBILITY CHECKLIST

We have made significant efforts to ensure the reproducibility of our work. All theoretical results are supported by clearly stated assumptions and complete proofs in the Appendix. The experimental setup, including dataset descriptions, preprocessing steps, and model hyperparameters, are detailed in Section X and Appendix Y. To facilitate replication, we provide an anonymized link to the source code and configuration files in the supplementary materials. Moreover, training logs, performance curves, and ablation studies are included to validate the reported results. Together, these resources allow independent researchers to reproduce both the theoretical and empirical findings of this paper.

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

## A APPENDIX

### A.1 PROOF OF LEMMA 1

Let $\mathcal{N} = \{1, 2, \ldots, N\}$ denote the set of devices, and $\tilde{g}_k^t = \frac{1}{N} \sum_{n \in \mathcal{N}} \tilde{g}_{n,k}^t$ the average of their local stochastic gradients at local iteration $t$ at global round $k$. We have

$$
-\mathbb{E}_{\{\xi_{1,k}^t, \ldots, \xi_{n,k}^t | \boldsymbol{\theta}_{1,k}^t, \ldots, \boldsymbol{\theta}_{N,k}^t\}} \mathbb{E}_{\{1,2,\ldots N\} \in \mathcal{N}} \left[ \langle \nabla f(\bar{\boldsymbol{\theta}}_k^t), \tilde{g}_k^t \rangle \right]
$$

$$
= -\mathbb{E}_{\{\xi_{1,k}^t, \ldots, \xi_{n,k}^t | \boldsymbol{\theta}_{1,k}^t, \ldots, \boldsymbol{\theta}_{N,k}^t\}}
$$
$$
\mathbb{E}_{\{1,2,\ldots N\} \in \mathcal{N}} \left[ \langle \nabla f(\bar{\boldsymbol{\theta}}_k^t), \frac{1}{N} \sum_{n \in \mathcal{N}} \tilde{g}_{n,k}^t \rangle \right]
$$

$$
\overset{\text{①}}{=} -\mathbb{E}_{\{1,2,\ldots N\} \in \mathcal{N}}
$$
$$
\mathbb{E}_{\{\xi_{1,k}^t, \ldots, \xi_{n,k}^t | \boldsymbol{\theta}_{1,k}^t, \ldots, \boldsymbol{\theta}_{N,k}^t\}} \left[ \langle \nabla f(\bar{\boldsymbol{\theta}}_k^t), \frac{1}{N} \sum_{n \in \mathcal{N}} \tilde{g}_{n,k}^t \rangle \right]
$$

$$
\overset{\text{②}}{=} \frac{1}{2} \Bigg[ -||\nabla f(\bar{\boldsymbol{\theta}}_k^t)||_2^2 - ||\sum_{n=0}^{N} \nabla f_n(\boldsymbol{\theta}_{n,k}^t)||_2^2 + ||\nabla f(\bar{\boldsymbol{\theta}}_k^t)
$$
$$
- \sum_{n=0}^{N} \nabla f_n(\boldsymbol{\theta}_{n,k}^t)||_2^2 \Bigg]
$$

$$
= \frac{1}{2} \Bigg[ -||\nabla f(\bar{\boldsymbol{\theta}}_k^t)||_2^2 - ||\sum_{n=0}^{N} \nabla f_n(\boldsymbol{\theta}_{n,k}^t)||_2^2
$$
$$
+ ||\sum_{n=0}^{N} \left( \nabla f_n(\bar{\boldsymbol{\theta}}_k^t) - \nabla f_n(\boldsymbol{\theta}_{n,k}^t) \right)||_2^2 \Bigg]
$$

$$
\overset{\text{③}}{\leq} \frac{1}{2} \Bigg[ -||\nabla f(\bar{\boldsymbol{\theta}}_k^t)||_2^2 - ||\sum_{n=0}^{N} \nabla f_n(\boldsymbol{\theta}_{n,k}^t)||_2^2
$$
$$
+ \sum_{n=0}^{N} ||\nabla f_n(\bar{\boldsymbol{\theta}}_k^t) - \nabla f_n(\boldsymbol{\theta}_{n,k}^t)||_2^2 \Bigg]
$$

$$
\overset{\text{④}}{\leq} \frac{1}{2} \Bigg[ -||\nabla f(\bar{\boldsymbol{\theta}}_k^t)||_2^2 - ||\sum_{n=0}^{N} \nabla f_n(\boldsymbol{\theta}_{n,k}^t)||_2^2
$$
$$
+ \sum_{n=0}^{N} L^2 ||\bar{\boldsymbol{\theta}}_k^t - \boldsymbol{\theta}_{n,k}^t||_2^2 + \eta_k \epsilon_{\text{BO}}^t + \frac{\eta_k L}{2} \Psi_k^t \Bigg], \tag{5}
$$

where ① is due to the fact that random variables $\xi_{n,k}^t$ and $\mathcal{N}$ are independent, ① is because ② $2\langle a, b \rangle = ||a||^2 + ||b||^2 - ||a - b||^2$, ③ holds due to the convexity of $||.||_2$, and ④ follows from Assumption 1, Assumption 4, and Assumption 5.

### A.2 PROOF OF LEMMA 2

We denote $k = i_c$ as the most recent global communication round, hence $\bar{\boldsymbol{\theta}}^{i_c+1} = \frac{1}{N} \sum_{n \in \mathcal{N}} \boldsymbol{\theta}_n^{i_c+1}$. The local solution at device $n$ at any particular iteration $i > i_c$, where $i$ is assumed to represent the most recent iteration, can be written as:

$$
\boldsymbol{\theta}_{n,k}^t = \boldsymbol{\theta}_n^i = \boldsymbol{\theta}_n^{i-1} - \eta_{i_c} \tilde{g}_n^{i-1} \overset{\text{①}}{=} \boldsymbol{\theta}_n^{i-2} - [\eta_{i_c} \tilde{g}_n^{i-2} + \eta_{i_c} \tilde{g}_n^{i-1}]
$$
$$
= \bar{\boldsymbol{\theta}}^{i_c+1} - \sum_{z=i_c+1}^{i-1} \eta_{i_c} \tilde{g}_n^z, \tag{6}
$$

where ② follows from the update rule of local solutions. Now, the average virtual model at iteration $i$ from equation 6 is computed as follows:

$$\bar{\boldsymbol{\theta}}^i = \bar{\boldsymbol{\theta}}^{i_c+1} - \frac{1}{N} \sum_{n \in \mathcal{N}} \sum_{z=i_c+1}^{i-1} \eta_{i_c} \tilde{g}_n^z. \tag{7}$$

Firstly, without loss of generality, suppose $i = s_t T + r$, with $s_t$ and $r$ denoting the indices of global communication round and local updates, respectively. Next, we consider that for $i_c + 1 < i \le i_c + T$, $\mathbb{E}_i || \bar{\boldsymbol{\theta}}^i - \boldsymbol{\theta}_n^i ||$ does not depend on time $i \le i_c$ for $1 \le n \le N$. Therefore, for all iterations $1 \le i \le I$, where $I = KT$, we can write,

$$\frac{1}{KT} \sum_{k=1}^{K} \sum_{t=1}^{T} \sum_{n=1}^{N} \mathbb{E} || \bar{\boldsymbol{\theta}}_k^t - \boldsymbol{\theta}_{n,k}^t ||^2 = \frac{1}{I} \sum_{i=1}^{I} \sum_{n=1}^{N} \mathbb{E} || \bar{\boldsymbol{\theta}}^i - \boldsymbol{\theta}_n^i ||^2$$

$$= \frac{1}{I} \sum_{s_t=1}^{\frac{I}{T}-1} \sum_{r=1}^{T} \sum_{n=1}^{N} \mathbb{E} || \bar{\boldsymbol{\theta}}^{s_t E + r} - \boldsymbol{\theta}_n^{s_t E + r} ||^2. \tag{8}$$

We bound the term $\mathbb{E} || \bar{\boldsymbol{\theta}}^i - \boldsymbol{\theta}_l^i ||^2$ for $i_c + 1 \le i = s_t T + r \le i_c + T$ in threes steps: (1) We begin by linking this quantity to the variance between the stochastic and full gradients, (2) Next, we invoke Assumption 1, which ensures unbiased estimation under i.i.d. mini-batch sampling. (3) Finally, we apply Assumption 3 to bound the final terms. We mention that $l$ is associated with individual client while $n$ is used for summing over devices.

*Relating to variance:*

$$\mathbb{E} || \bar{\boldsymbol{\theta}}^{s_t E + r} - \boldsymbol{\theta}_l^{s_t E + r} ||^2$$

$$= \mathbb{E} || \bar{\boldsymbol{\theta}}^{i_c+1} - \left[ \sum_{z=i_c+1}^{i-1} \eta_{i_c} \tilde{g}_l^z \right] - \bar{\boldsymbol{\theta}}^{i_c+1}$$

$$+ \left[ \frac{1}{N} \sum_{n \in \mathcal{N}} \sum_{z=i_c+1}^{i-1} \eta_{i_c} \tilde{g}_n^z \right] ||^2$$

$$\stackrel{①}{=} \mathbb{E} || \sum_{z=1}^{r} \eta_{i_c} \tilde{g}_l^{s_t+z} - \frac{1}{N} \sum_{n \in \mathcal{N}} \sum_{z=1}^{r} \eta_{i_c} \tilde{g}_n^{s_t+z} ||^2$$

$$\stackrel{②}{\le} 2 \left[ \mathbb{E} || \sum_{z=1}^{r} \eta_{i_c} \tilde{g}_l^{s_t+z} ||^2 - \mathbb{E} || \frac{1}{N} \sum_{n \in \mathcal{N}} \sum_{z=1}^{r} \eta_{i_c} \tilde{g}_n^{s_t+z} ||^2 \right]$$

$$\stackrel{③}{=} 2 \left[ \mathbb{E} || \sum_{z=1}^{r} \eta_{i_c} \tilde{g}_l^{s_t+z} - \mathbb{E} \left[ \sum_{z=1}^{r} \eta_{i_c} \tilde{g}_l^{s_t+z} \right] ||^2 \right.$$

$$- \mathbb{E} || \sum_{z=1}^{r} \eta_{i_c} \tilde{g}_l^{s_t+z} ||^2 + \mathbb{E} || \frac{1}{N} \sum_{n \in \mathcal{N}} \sum_{z=1}^{r} \eta_{i_c} \tilde{g}_n^{s_t+z}$$

$$- \mathbb{E} \left[ \frac{1}{N} \sum_{n \in \mathcal{N}} \sum_{z=1}^{r} \eta_{i_c} \tilde{g}_n^{s_t+z} \right] ||^2 + || \mathbb{E} \left[ \frac{1}{N} \sum_{n \in \mathcal{N}} \sum_{z=1}^{r} \eta_{i_c} \tilde{g}_n^{s_t+z} \right] ||^2$$

$$\stackrel{④}{=} 2\mathbb{E} \left( \left[ || \sum_{z=1}^{r} \eta_{i_c} \left[ \tilde{g}_l^{s_t T + z} - g_l^{s_t T + z} \right] ||^2 \right. \right.$$

$$+ || \sum_{z=1}^{r} \eta_{i_c} g_l^{s_t T + z} ||^2 \right]$$

$$+ || \frac{1}{N} \sum_{n \in \mathcal{N}} \sum_{z=1}^{r} \eta_{i_c} \left[ \tilde{g}_n^{s_t T + z} - g_n^{s_t T + z} \right] ||^2$$

$$+ || \frac{1}{N} \sum_{n \in \mathcal{N}} \sum_{z=1}^{r} \eta_{i_c} g_n^{s_t T + z} ||^2 \right),$$

where ① holds because $i = s_t T + r \leq i_c + T$, ② is due to $||a - b||^2 \leq 2(||a||^2 + ||b||^2)$, ③ arises because of $\mathbb{E}[\boldsymbol{\theta}^2] = \mathbb{E}[[\boldsymbol{\theta} - \mathbb{E}[\boldsymbol{\theta}]]^2] + \mathbb{E}[\boldsymbol{\theta}]^2$, ④ comes from Assumption 1.

*Unbiased estimation and i.i.d. sampling*

$$
\overset{⑤}{=} 2\mathbb{E}\Bigg( \Bigg[ \sum_{z=1}^{r} \eta_{i_c}^2 ||\tilde{g}_l^{s_t T + z} - g_l^{s_t T + z}||^2 + || \sum_{z=1}^{r} \eta_{i_c} g_l^{s_t T + z}||^2 \Bigg]
$$

$$
+ \frac{1}{N^2} \sum_{n \in \mathcal{N}} \sum_{z=1}^{r} \eta_{i_c}^2 ||\tilde{g}_n^{s_t T + z} - g_n^{s_t T + z}||^2
$$

$$
+ ||\frac{1}{N} \sum_{n \in \mathcal{N}} \sum_{z=1}^{r} \eta_{i_c} g_n^{s_t T + z}||^2 \Bigg)
$$

$$
\overset{⑥}{\leq} 2\mathbb{E}\Bigg( \Bigg[ \sum_{z=1}^{r} \eta_{i_c}^2 ||\tilde{g}_l^{s_t T + z} - g_l^{s_t T + z}||^2 + r \sum_{z=1}^{r} \eta_{i_c}^2 ||g_l^{s_t T + z}||^2 \Bigg]
$$

$$
+ \frac{1}{N^2} \sum_{n \in \mathcal{N}} \sum_{z=1}^{r} ||\tilde{g}_n^{s_t T + z} - g_n^{s_t T + z}||^2
$$

$$
+ \frac{r}{N^2} \sum_{n \in \mathcal{N}} \sum_{z=1}^{r} \eta_{s_t T + z}^2 ||g_n^{s_t T + z}||^2
$$

$$
= 2\Bigg( \Bigg[ \sum_{z=1}^{r} \eta_{i_c}^2 \mathbb{E}||\tilde{g}_l^{s_t T + z} - g_l^{s_t T + z}||^2 + r \sum_{z=1}^{r} \eta_{i_c}^2 \mathbb{E}||g_l^{s_t T + z}||^2 \Bigg]
$$

$$
+ \frac{1}{N^2} \sum_{n \in \mathcal{N}} \sum_{z=1}^{r} \eta_{i_c}^2 \mathbb{E}||\tilde{g}_n^{s_t T + z} - g_n^{s_t T + z}||^2
$$

$$
+ \frac{r}{N^2} \sum_{n \in \mathcal{N}} \sum_{z=1}^{r} \eta_{i_c}^2 \mathbb{E}||g_n^{s_t T + z}||^2 \Bigg), \tag{9}
$$

where ⑤ is due to independent mini-batch sampling as well as unbiased estimation assumption, and ⑥ follows from the inequality $|| \sum_{i=1}^{m} a_i||^2 \leq m \sum_{i=1}^{m} ||a_i||^2$.

*Using Assumption 3:* Our next step is to bound the terms in equation 9 using Assumption 3 as follows:

$$
\mathbb{E}||\bar{\boldsymbol{\theta}}_k^t - \boldsymbol{\theta}_{l,k}^t||^2 \leq 2\Bigg( \Bigg[ \sum_{z=1}^{r} \eta_{i_c}^2 \Big[ C_1 ||g_l^{s_t T + z}||^2 + \frac{\sigma^2}{B} \Big]
$$

$$
+ r \sum_{z=1}^{r} \eta_{i_c}^2 ||g_l^{s_t T + z}||^2 + \frac{1}{N^2} \sum_{n \in \mathcal{N}} \sum_{z=1}^{r} \eta_{i_c}^2 \Big[ C_1 ||g_n^{s_t T + z}||^2 + \frac{\sigma^2}{B} \Big]
$$

$$
+ \frac{r}{N^2} \sum_{n \in \mathcal{N}} \sum_{z=1}^{r} \eta_{i_c}^2 ||g_n^{s_t T + z}||^2 \Bigg)
$$

$$
= 2\Bigg( \Bigg[ \sum_{z=1}^{r} \eta_{i_c}^2 C_1 ||g_l^{s_t T + z}||^2 + \sum_{z=1}^{r} \eta_{i_c}^2 \frac{\sigma^2}{B}
$$

$$
+ r \sum_{z=1}^{r} \eta_{i_c}^2 ||g_l^{s_t T + z}||^2 \Bigg] + \frac{1}{N^2} \sum_{n \in \mathcal{N}} \sum_{z=1}^{r} \eta_{i_c}^2 C_1 ||g_n^{s_t T + z}||^2
$$

$$
+ \sum_{z=1}^{r} \eta_{i_c}^2 \frac{\sigma^2}{NB} + \frac{r}{N^2} \sum_{n \in \mathcal{N}} \sum_{z=1}^{r} \eta_{i_c}^2 ||g_n^{s_t T + z}||^2 \Bigg).
$$

$$
\tag{10}
$$

Now we determine the upper bound for $\sum_{r=1}^{T}\sum_{n=1}^{N}[\mathbb{E}||\bar{\boldsymbol{\theta}}_k^t - \boldsymbol{\theta}_{n,k}^t||]$ using equation 10 as follows:

$$
\sum_{r=1}^{T}\sum_{n=1}^{N}\left[\mathbb{E}||\bar{\boldsymbol{\theta}}^{s_t T+z} - \boldsymbol{\theta}_n^{s_t T+z}||\right]
$$

$$
\leq 2\sum_{r=1}^{T}\sum_{l=1}^{N}\left(\left[\sum_{z=1}^{r}\eta_{i_c}^2 C_1||g_l^{s_t T+z}||^2 + \sum_{z=1}^{r}\eta_{i_c}^2\frac{\sigma^2}{B}\right.\right.
$$

$$
\left. + r\sum_{z=1}^{r}\eta_{i_c}^2||g_l^{s_t T+z}||^2\right] + \frac{1}{N^2}\sum_{n\in\mathcal{N}}\sum_{z=1}^{r}\eta_{i_c}^2 C_1||g_n^{s_t T+z}||^2
$$

$$
\left. + \sum_{z=1}^{r}\eta_{i_c}^2\frac{\sigma^2}{NB} + \frac{r}{N^2}\sum_{n\in\mathcal{N}}\sum_{z=1}^{r}\eta_{i_c}^2||g_n^{s_t T+z}||^2\right)
$$

$$
\overset{\textcircled{1}}{\leq} 2\eta_{i_c}^2\left(\left[\sum_{z=1}^{T}C_1\sum_{l=1}^{N}||g_l^{s_t T+z}||^2 + \frac{T(T+1)\sigma^2}{2B}\right.\right.
$$

$$
+ \frac{T(T+1)}{2}\sum_{z=1}^{T}\sum_{l=1}^{N}||g_l^{s_t T+z}||^2
$$

$$
+ \frac{1}{N^2}\sum_{n\in\mathcal{N}}\sum_{z=1}^{T}C_1||g_n^{s_t T+z}||^2
$$

$$
\left. + \frac{T(T+1)\sigma^2}{2NB} + \frac{T(T+1)}{2N^2}\sum_{n\in\mathcal{N}}\sum_{z=1}^{T}||g_n^{s_t T+z}||^2\right)
$$

$$
= \frac{\eta_{i_c}^2(N+1)}{N}\left(\left[(2C_1 + T(T+1))\sum_{z=1}^{T}\sum_{n=1}^{N}||g_n^{s_t T+z}||^2\right]\right.
$$

$$
\left. + \frac{T(T+1)\sigma^2}{B}\right), \tag{11}
$$

where ① follows from the fact that the terms $||g_l||^2$ are positive. Now, taking summation over global communication rounds in equation 11 gives:

$$
\sum_{s_t=1}^{I/T-1}\sum_{r=1}^{T}\sum_{n=1}^{N}\left[\mathbb{E}||\bar{\boldsymbol{\theta}}^{s_t T+z} - \boldsymbol{\theta}_n^{s_t T+z}||\right]
$$

$$
\leq \frac{\eta_{i_c}^2(N+1)}{N}\left(\left[(2C_1\right.\right.
$$

$$
+ T(T+1))\sum_{s_t=1}^{I/T-1}\sum_{z=1}^{T}\sum_{n=1}^{N}||g_n^{s_t T+z}||^2\right]
$$

$$
\left. + \frac{I(T+1)\sigma^2}{B}\right)
$$

$$
= \frac{\eta_{i_c}^2(N+1)}{N}\left(\left[(2C_1 + T(T+1))\sum_{i=1}^{I}\sum_{n=1}^{N}||g_n^i||^2\right]\right.
$$

$$
\left. + \frac{I(T+1)\sigma^2}{B}\right), \tag{12}
$$

which leads to

$$\frac{1}{I} \sum_{i=1}^{I} \sum_{n=1}^{N} \left[ \mathbb{E} \| \bar{\boldsymbol{\theta}}^i - \boldsymbol{\theta}_n^i \| \right]$$

$$\leq \frac{(2C_1 + T(T+1))}{I} \frac{\eta_{i_c}^2 (N+1)}{N} \sum_{i=0}^{I-1} \sum_{n=1}^{N} \| g_n^i \|^2$$

$$+ \frac{\eta_{i_c}^2 I (N+1)(T+1)\sigma^2}{NB}$$

$$\overset{①}{\leq} \frac{(2C_1 + T(T+1))}{I} \frac{\lambda \eta_{i_c}^2 (N+1)}{N} \sum_{i=0}^{I-1} \| \sum_{n=1}^{N} g_n^i \|^2$$

$$+ \frac{\eta_{i_c}^2 I (N+1)(T+1)\sigma^2}{NB}, \tag{13}$$

where ① follows from the definition of weighted gradient diversity and upper bound assumption in (30) of the main paper. Finally, equation 13 can be written as:

$$\frac{1}{KT} \sum_{k=1}^{K} \sum_{t=1}^{T} \sum_{n=1}^{N} \left[ \mathbb{E} \| \bar{\boldsymbol{\theta}}_k^t - \boldsymbol{\theta}_{n,k}^t \| \right]$$

$$\leq \frac{(2C_1 + T(T+1))}{KT} \frac{\lambda \eta_{i_c}^2 (N+1)}{N} \sum_{k=1}^{K} \sum_{t=1}^{T} \| \sum_{n=1}^{N} g_{n,k}^t \|^2$$

$$+ \frac{\eta_{i_c}^2 KT (N+1)(T+1)\sigma^2}{NB}. \tag{14}$$

### A.3 PROOF OF LEMMA 3

We have

$$\mathbb{E} \left[ \| \tilde{g}_k^t - g_k^t \|^2 \right] \overset{①}{=} \mathbb{E} \left[ \| \frac{1}{N} \sum_{n=0}^{N} \tilde{g}_{n,k}^t - g_{n,k}^t \|^2 \right]$$

$$= \frac{1}{N^2} \mathbb{E} \left[ \sum_{n=0}^{N} \| (\tilde{g}_{n,k}^t - g_{n,k}^t) \|^2 \right]$$

$$+ \sum_{i \neq n} \langle \tilde{g}_{i,k}^t - g_{i,k}^t, \tilde{g}_{n,k}^t - g_{n,k}^t \rangle$$

$$= \frac{1}{N^2} \sum_{n=0}^{N} \mathbb{E} \| (\tilde{g}_{n,k}^t - g_{n,k}^t) \|^2$$

$$+ \sum_{i \neq n} \frac{1}{N^2} \mathbb{E} \left[ \langle \tilde{g}_{i,k}^t - g_{i,k}^t, \tilde{g}_{n,k}^t - g_{n,k}^t \rangle \right]$$

$$\overset{②}{=} \frac{1}{N^2} \sum_{n=0}^{N} \mathbb{E} \| (\tilde{g}_{n,k}^t - g_{n,k}^t) \|^2$$

$$+ \frac{1}{N^2} \sum_{i \neq n} \langle \mathbb{E} \left[ \tilde{g}_{i,k}^t - g_{i,k}^t \right], \mathbb{E} \left[ \tilde{g}_{n,k}^t - g_{n,k}^t \right] \rangle$$

$$\overset{③}{\leq} \frac{1}{N^2} \sum_{n=0}^{N} \left[ C_1 \| g_{n,k}^t \|^2 + C_2^2 \right] = \frac{C_1}{N^2} \sum_{n=0}^{N} \| g_{n,k}^t \|^2 + \frac{C_2^2}{N}, \tag{15}$$

where we use the definition of $\tilde{g}_k^t$ and $g_k^t$ in ①, in ② we use the fact that mini-batches are chosen in i.i.d. manner at each device, and ③ follows directly from Assumption 3. We note that Assumption

3 implies $\mathbb{E}[\tilde{g}_{n,k}^t] = g_{n,k}^t$. Therefore. we have

$$
\mathbb{E}\left[||\tilde{g}_k^t||^2\right] = \mathbb{E}\left[||\tilde{g}_k^t - \mathbb{E}[\tilde{g}_k^t]||^2\right] + ||\mathbb{E}[\tilde{g}_k^t]||^2
$$

$$
= \mathbb{E}\left[||\tilde{g}_k^t - g_k^t||^2\right] + ||g_k^t||^2
$$

$$
\overset{①}{\leq} \frac{C_1}{N^2}\sum_{n=0}^N ||g_{n,k}^t||^2 + \frac{C_2^2}{N} + ||\frac{1}{N}\sum_{n=0}^N g_{n,k}^t||^2
$$

$$
\overset{②}{\leq} \frac{C_1}{N^2}\sum_{n=0}^N ||g_{n,k}^t||^2 + \frac{C_2^2}{N} + \frac{1}{N}\sum_{n=0}^N ||g_{n,k}^t||^2 + L^2\psi_k^t
$$

$$
= \left(\frac{C_1+N}{N^2}\right)\sum_{n=0}^N ||g_{n,k}^t||^2 + \frac{C_2^2}{N} + L^2\psi_k^t, \tag{16}
$$

where ① and ② follows from the fact that $||\sum_{i=1}^m a_i||^2 \leq m\sum_{i=1}^m ||a_i||^2$, with $a_i \in \mathbb{R}^n$, and Assumption 4 and Assumption 5. Using the upper bound over the weighted gradient diversity, $\lambda$,

$$
\mathbb{E}\left[||\tilde{g}_k^t||^2\right] \leq \lambda\left(\frac{C_1+N}{N^2}\right)||\sum_{n=0}^N g_{n,k}^t||^2 + \frac{C_2^2}{N} + L^2\psi_k^t, \tag{17}
$$

results in the stated bound.

## A.4 PROOF OF LEMMA 4

To prove Lemma 4, we fix the indices related to global and local iteration $k$ and $t$, consequently dropping them from notations temporarily. Let $X_{n,d,\pm} = \langle\hat{Z}\rangle_{|\Psi_n(\boldsymbol{\theta}_n \pm \frac{\pi}{2}e_d)\rangle} - \langle Z\rangle_{|\Psi_n(\boldsymbol{\theta}_n \pm \frac{\pi}{2}e_d)\rangle}$ denote the difference between the estimated and true expectation of the observable $Z$ under the quantum state $|\Psi_n(\boldsymbol{\theta}_n \pm \frac{\pi}{2}e_d)\rangle$ whose $d^{\text{th}}$ parameter is phase shifted by $\pm\frac{\pi}{2}$. In the following analysis, we use the notation $|\Psi_{n,d,\pm}\rangle = |\Psi_n(\boldsymbol{\theta}_n \pm \frac{\pi}{2}e_d)\rangle$ for brevity. The variance of the gradient estimate in equation 30 is written as

$$
\text{var}(\xi_n) = \mathbb{E}\Bigg[\sum_{d=1}^D \Big(\frac{1}{2}(\langle\hat{Z}\rangle_{|\Psi_{n,d,+}\rangle} - \langle\hat{Z}\rangle_{|\Psi_{n,d,-}\rangle})
$$

$$
- \frac{1}{2}(\langle Z\rangle_{|\Psi_{n,d,+}\rangle} - \langle Z\rangle_{|\Psi_{n,d,-}\rangle})\Big)^2\Bigg]
$$

$$
= \sum_{d=1}^D \frac{1}{4}\mathbb{E}\left[\left(X_{n,d,+} - X_{n,d,-}\right)^2\right]
$$

$$
= \sum_{d=1}^D \frac{1}{4}\left(\mathbb{E}[X_{n,d,+}^2] - \mathbb{E}[X_{n,d,-}^2]\right), \tag{18}
$$

where the expectation is taken with respect to the $H$ measurements of the quantum states $|\Psi_n(\boldsymbol{\theta}_n + \frac{\pi}{2}e_d)\rangle$ and $|\Psi_n(\boldsymbol{\theta}_n - \frac{\pi}{2}e_d)\rangle$ for $d = 1, 2, \ldots, D$. Hence, the random variables $X_{n,d,+}$ and $X_{n,d,-}$ are independent for $d = 1, 2, \ldots, D$, which results in the equality in equation 18. It is to note that the expectation $\mathbb{E}[X_{n,d,+}^2]$ is equal to the variance $\text{var}(\langle\hat{Z}\rangle_{|\Psi_{n,d,+}\rangle})$ of the random variable $\langle\hat{Z}\rangle_{|\Psi_{n,d,+}\rangle}$. Let $Y$ be the random variable that defines the index of the measurement of the observable $Z$. Therefore, $Z = h_Y$ represents the corresponding measurement output. We denote the Bernoulli random variable as $W_y = \mathbb{I}\{Y = y\}$ determining whether $Y = y(W_y = 1)$ or not ($W_y = 0$). We also mention that the quantum measurements are i.i.d., and thus it follows from the

definition of expectation of $\langle \hat{Z} \rangle_{|\Psi_{n,d,+}\rangle}$ that

$$
\begin{aligned}
\mathbb{E}[X_{n,d,+}^2] &= \frac{1}{H}\operatorname{var}\left(\sum_{y=1}^{N_z} h_y W_y\right) \\
&= \frac{1}{H}\mathbb{E}\left[\left(\sum_{y=1}^{N_z} h_y(W_y - p(y|\boldsymbol{\theta}_n + e_d\frac{\pi}{2}))\right)^2\right] \\
&\overset{\text{\textcircled{1}}}{\leq} \frac{1}{H}\left(\sum_{y=1}^{N_z} h_y^2\right)\sum_{y=1}^{N_z}\operatorname{var}(W_y) \\
&\overset{\text{\textcircled{2}}}{=} \frac{1}{H}\left(\sum_{y=1}^{N_z} h_y^2\right)\sum_{y=1}^{N_z} v\left(p(y|\boldsymbol{\theta}_n + e_d\frac{\pi}{2})\right) \\
&\leq \frac{N_z}{N_y}\left(\sum_{y=1}^{N_z} h_y^2\right)v = \frac{N_z \operatorname{Tr}(Z^2)}{H}v,
\end{aligned}
\tag{19}
$$

where ① follows from the Cauchy-Schwarz inequality, ② is due to the fact that the variance of the Bernoulli random variable $W_y$ is computed as

$$
\operatorname{var}\left(W_y\right) = \mathbb{E}\left[W_y^2\right] - \left(\mathbb{E}\left[W_y\right]\right)^2 = v\left(p(y|\boldsymbol{\theta}_n + e_d\frac{\pi}{2})\right),
\tag{20}
$$

where $v(x) = x(1-x)$ for $x \in (0,1)$. The last yields from the definition of the quantity $v$. In a similar way, it can be shown that the following inequality holds

$$
\mathbb{E}[X_{n,d,-}^2] \leq \frac{N_z \operatorname{Tr}(Z^2)}{H}v.
\tag{21}
$$

From equation 19 and equation 21, we can write while bringing the omitted indices back

$$
\operatorname{var}(\xi_{n,k}^t) \leq \frac{\nu N_z D Tr(Z^2)}{2H}.
\tag{22}
$$

For $N$ number of QFL clients, we get

$$
\operatorname{var}(\xi_k^t) \leq \frac{1}{N}\sum_{n \in \mathcal{N}} \frac{\nu N_z D Tr(Z^2)}{2H},
\tag{23}
$$

concluding the proof.

### A.5 PROOF OF THEOREM 1

Using Lemma 1 and Lemma 2, we continue to further upper bound (34) of main paper as follows:

$$
\frac{1}{KT} \sum_{k=1}^{K} \sum_{t=1}^{T} \mathbb{E}[f(\bar{\boldsymbol{\theta}}_k^{t+1}) - f(\bar{\boldsymbol{\theta}}_k^t)]
$$

$$
\leq \frac{1}{KT} \sum_{k=1}^{K} \sum_{t=1}^{T} \left( -\eta_k \mathbb{E}\left[ \langle \nabla f(\bar{\boldsymbol{\theta}}_k^t), \tilde{g}_k^t \rangle \right] \right)
$$

$$
+ \frac{1}{KT} \sum_{k=1}^{K} \sum_{t=1}^{T} \frac{\eta_k^2 L}{2} \mathbb{E}\left[ ||\tilde{g}_k^t||^2 \right]
$$

$$
\leq \frac{1}{KT} \sum_{k=1}^{K} \sum_{t=1}^{T} \left( -\frac{\eta_k}{2} ||\nabla f(\bar{\boldsymbol{\theta}}_k^t)||^2 - \frac{\eta_k}{2} ||\sum_{n=1}^{N} \nabla f_n(\boldsymbol{\theta}_{n,k}^t)||^2 \right)
$$

$$
+ \frac{\lambda \eta_k L^2}{2KT} \frac{N+1}{N} \left( \left[ 2C_1 + T(T+1) \right] \eta_k^2 \frac{1}{KT} \sum_{k=1}^{K} \sum_{t=1}^{T} ||^2 \right.
$$

$$
- \frac{\eta_k}{2} ||\sum_{n=1}^{N} \nabla f_n(\boldsymbol{\theta}_{n,k}^t)||^2 \Big)
$$

$$
+ \frac{\eta_k L^2}{2KT} \left( \frac{N+1}{N} \right) \left( \frac{KT(T+1)\eta_k^2 \sigma^2}{B} \right)
$$

$$
+ \frac{1}{KT} \sum_{k=1}^{K} \sum_{t=1}^{T} \frac{L\eta_k^2}{2} \left( \lambda \left( \frac{C_1}{N} + 1 \right) \left[ ||\sum_{n=1}^{N} \nabla f_n(\boldsymbol{\theta}_{n,k}^t)||^2 \right] \right.
$$

$$
+ \frac{\sigma^2}{NB} \Big) + L\psi_k^t
$$

$$
= \frac{1}{KT} \sum_{k=1}^{K} \sum_{t=1}^{T} \left( -\frac{\eta_k}{2} ||\nabla f(\bar{\boldsymbol{\theta}}_k^t)||^2 - \frac{\eta_k}{2} ||\sum_{n=1}^{N} \nabla f_n(\boldsymbol{\theta}_{n,k}^t)||^2 \right)
$$

$$
+ \frac{\lambda \eta_k L^2}{2KT} \frac{N+1}{N} \left( \lambda \left[ 2C_1 + T(T+1) \right] \eta_k^2 \frac{1}{KT} \sum_{k=1}^{K} \sum_{t=1}^{T} ||^2 \right.
$$

$$
- \frac{\eta_k}{2} ||\sum_{n=1}^{N} \nabla f_n(\boldsymbol{\theta}_{n,k}^t)||^2 \Big)
$$

$$
+ \frac{KT(L+1)\eta_k^2 \sigma^2}{B} + \frac{1}{KT} \sum_{k=1}^{K} \sum_{t=1}^{T} \frac{\lambda L\eta_k^2}{2} \lambda \left( \frac{C_1}{N} + 1 \right)
$$

$$
\left[ ||\sum_{n=1}^{N} \nabla f_n(\boldsymbol{\theta}_{n,k}^t)||^2 \right] + \frac{L\eta_k^2}{2} \frac{\sigma^2}{NB} + L\psi_k^t. \tag{24}
$$

From equation 24, we have

$$
\frac{1}{KT} \sum_{k=1}^{K} \sum_{t=1}^{T} \mathbb{E}[f(\bar{\boldsymbol{\theta}}_k^{t+1}) - f(\bar{\boldsymbol{\theta}}_k^t)]
$$

$$
\leq -\frac{1}{KT} \sum_{k=1}^{K} \sum_{t=1}^{T} \frac{\eta_k}{2} ||\nabla f(\bar{\boldsymbol{\theta}}_k^t)||^2
$$

$$
\tag{25}
$$

$$+ \frac{1}{KT} \sum_{k=1}^{K} \sum_{t=1}^{T} \left[ - \frac{\eta_k}{2} + \frac{\lambda(N+1)L^2\eta_k^3[2C_1 + T(T+1)]}{2N} \right.$$

$$\left. + \frac{\lambda L \eta_k^2}{2} \left( \frac{C_1}{N} + 1 \right) \right] \left[ \| \sum_{n=1}^{N} \nabla f_n(\boldsymbol{\theta}_{n,k}^t) \|^2 \right]$$

$$+ \frac{\eta_k^3 L^2 (T+1) \sigma^2}{B} \left( \frac{N+1}{N} \right) + \frac{L\eta_k^2}{2} \frac{\sigma^2}{NB} + \frac{L^2\eta_k^2}{2KT} \sum_{k=1}^{K} \sum_{t=1}^{T} \psi_k^t$$

$$\overset{\textcircled{1}}{\leq} - \frac{1}{KT} \sum_{k=1}^{K} \sum_{t=1}^{T} \frac{\eta_k}{2} \| \nabla f(\bar{\boldsymbol{\theta}}_k^t) \|^2$$

$$+ \frac{\eta_k^3 L^2 (T+1) \sigma^2}{B} \left( \frac{N+1}{N} \right) + \frac{L\eta_k^2}{2} \frac{\sigma^2}{NB} + \frac{L^2\eta_k^2}{2KT} \sum_{k=1}^{K} \sum_{t=1}^{T} \psi_k^t, \tag{26}$$

where $\textcircled{1}$ follows if the following condition holds:

$$- \frac{\eta_k}{2} + \frac{\lambda(N+1)L^2\eta_k^3[2C_1 + T(T+1)]}{2N}$$

$$+ \frac{\lambda L \eta_k^2}{2} \left( \frac{C_1}{N} + 1 \right) \leq 0. \tag{27}$$

In any kind of FL framework, setting the coefficient of the local gradients' sum to zero helps control variance from diverse client updates, ensuring stable convergence. This condition limits the influence of individual clients on the global model, preventing oscillations or divergence. It keeps updates bounded, promoting reliable convergence toward an optimal solution. By rearranging equation 26, we get

$$\frac{1}{KT} \sum_{k=1}^{K} \sum_{t=1}^{T} \mathbb{E} \| \nabla f(\bar{\boldsymbol{\theta}}_k^t) \|^2 \leq \frac{2[f(\bar{\boldsymbol{\theta}}_1^0) - f^*]}{\eta_k KT} + \frac{L\eta\sigma^2}{NB}$$

$$+ \frac{2\eta_k^2 \sigma^2 L^2 (T+1)}{B} + \frac{2}{KT} \sum_{k,t} \epsilon_{\text{BO}}^t + \frac{L}{KT} \sum_{k,t} \Psi_k^t. \tag{28}$$

Upto this point, we did not consider noise term in the local gradient of the quantum client. However, we have to consider that because the noise term in the local gradient of each quantum device will affect the convergence of the overall global gradient. Since the global gradient in QFL is an aggregation of the local gradients from all the devices, any noise or error in the local gradient estimates will also accumulate at the global level. Hence, we find the upper bound of the variance of the error introduced in the local gradient of each client due to quantum shot noise and add it to the upper bound of the global gradient in equation 28.

In QFL, the gradient is estimated rather than explicitly computed. This approach leverages quantum computations to approximate the gradient, allowing for efficient optimization processes without relying on exact gradient calculations. Our assumption $\mathbb{E}[\tilde{g}_n] = g_n$ means that the estimate is unbiased. Therefore, we can write

$$\tilde{g}_{n,k}^t = g_{n,k}^t + \xi_{n,k}^t, \tag{29}$$

where $\tilde{g}_{n,k}^t$ is the stochastic estimate of the gradient, $g_{n,k}^t$ is the true gradient, and $\xi_{n,k}^t$ is the error or noise introduced in the estimation process, with the noise term satifying the conditions $\mathbb{E}[\xi_{n,k}^t] = 0$ and $\text{var}(\xi_{n,k}^t) = \mathbb{E}[\|\tilde{g}_{n,k}^t - g_{n,k}^t\|]$ Taking average across all the devices, we get

$$\tilde{g}_k^t = g_k^t + \xi_k^t, \tag{30}$$

where $\xi_k^t = \sum_{n=0}^{N} \xi_{n,k}^t$. Since the global gradient in QFL is an aggregation of the local gradients from all the devices, any noise or error in the local gradient estimates will also accumulate at the global level. If the errors are significant, they may cause the aggregated global gradient to deviate from the true direction of descent, slowing down convergence or leading to suboptimal solutions.

Hence, we use Lemma 4 to find the upper bound of the variance of the gradient estimate. Therefore, The expected value of the squared norm of the global gradient in equation 28 will be additionally bounded by the left hand side of Lemma 4 in the following way:

$$
\frac{1}{KT} \sum_{k=1}^{K} \sum_{t=1}^{T} \mathbb{E}||\nabla f(\bar{\boldsymbol{\theta}}_k^t)||^2 \leq \frac{2[f(\bar{\boldsymbol{\theta}}_1^0) - f^*]}{\eta_k KT} + \frac{L\eta\sigma^2}{NB}
$$

$$
+ \frac{2\eta_k^2 \sigma^2 L^2 (T+1)}{B} \left(\frac{N+1}{N}\right) + \frac{1}{N} \sum_{n \in \mathcal{N}} \frac{\nu N_z D Tr(Z^2)}{2H}
$$

$$
+ \frac{2}{KT} \sum_{k,t} \epsilon_{\text{BO}}^t + \frac{L}{KT} \sum_{k,t} \Psi_k^t. \tag{31}
$$

In non-convex optimization, achieving a global minimum is often infeasible due to the landscape's complexity, filled with local minima and saddle points. Instead of focusing on bounding the distance between consecutive points, an alternative approach is to bound the squared norm of the gradient estimate. This approach helps gauge how close we are to a stationary point, where the gradient's magnitude is minimal, indicating minimal change. By upper bounding the squared gradient, we can evaluate convergence towards a solution that may not be globally optimal, however is practically effective in reducing the loss.

### A.6 PERFORMANCE OF QFL FRAMEWORK

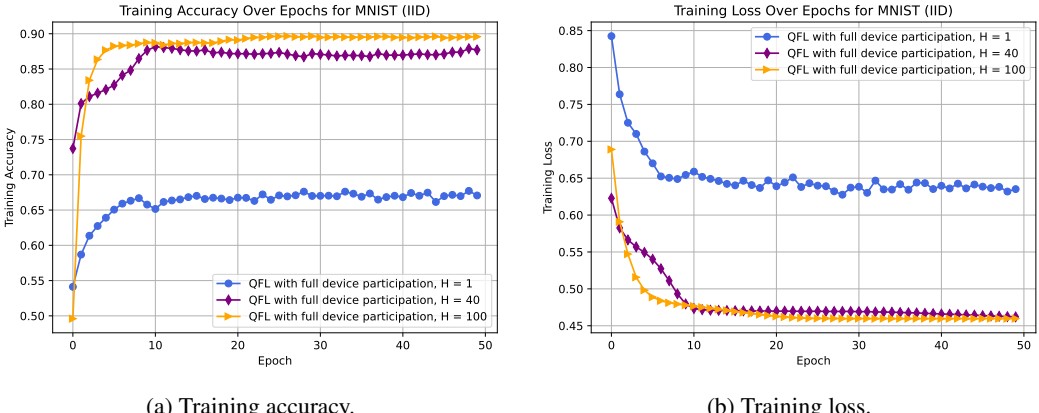

(a) Training accuracy.

(b) Training loss.

Figure 5: Training performance comparison of QFL for MNIST dataset (IID) for full device participation with varying number of quantum measurement shots.

Fig. 5 illustrates the training performance in QFL on MNIST dataset (IID), showcasing improved accuracy with increasing number of quantum measurement shots under full device participation scenario. Figures 5a and 5b show that increasing the number of quantum measurement shots ($H$=1, $H$=40, $H$=100) significantly improves QFL performance. Higher shot counts reduce quantum shot noise by averaging more measurement outcomes, leading to greater accuracy and lower loss. Moving from $H$=1 to $H$=40 and then $H$=100 consistently enhances stability and reliability, highlighting the importance of scaling up measurement shots for robust training in QFL systems. Fig. 6 shows that under the non-IID MNIST setting, increasing quantum measurement shots yields the same trend as in the IID case (Fig. 5), consistently improving accuracy and reducing loss regardless of data distribution.

### A.7 DETAILS ON DATA HETEROGENEITY

As an example, Figure 7 illustrates the extreme non-IID label distribution for the MNIST dataset across three quantum clients. The 3D plot encodes class index (digits 0–9) on the x-axis, client ID on the y-axis, and the per-class sample count on the z-axis. Each client curve with shaded underlay represents the number of samples available for each digit in that client's local dataset. In this setup,

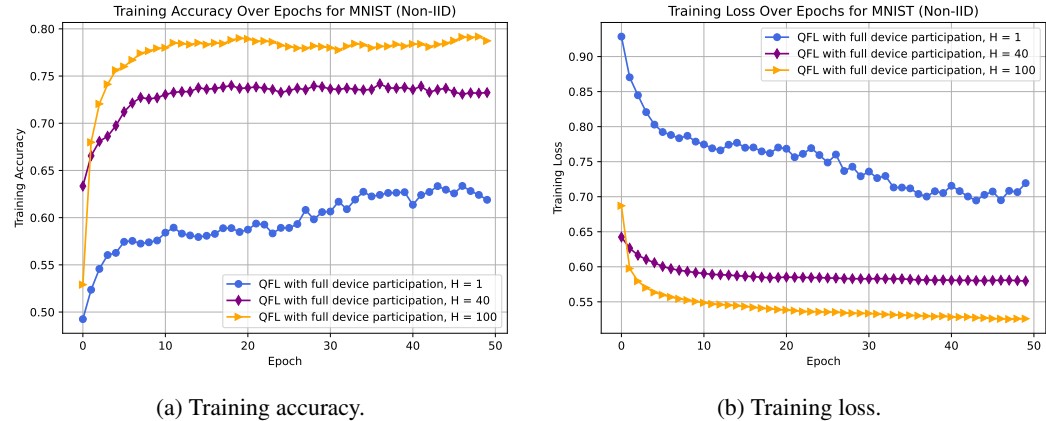

(a) Training accuracy.

(b) Training loss.

Figure 6: Training performance comparison of QFL on Cifar10 and MNIST datasets (non-IID) for full device participation with varying number of quantum measurement shots.

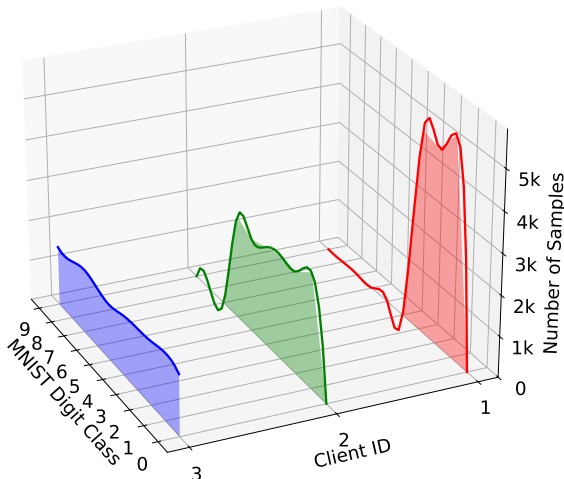

Figure 7: Heterogeneous distribution of MNIST data among quantum clients. Each client has an imbalanced class distribution and a varying number of data instances compared to other clients.

Client 1 is restricted to digits 1, 2, 3 with close to 18 thousand samples, Client 2 to 1, 2, 3, 4, 5, 6 with close to 14 thousand samples, and Client 3 to all digits 0–9 with close to 13 thousand samples. This configuration introduces both label-support mismatch and sample imbalance, ultimately creating a challenging heterogeneous scenario for QFL with clients to see varying subsets and quantities of labels.

Figure 8 shows the analogous setup for the HAR dataset. Here, Client 1 is restricted to activities Laying, Standing with close to 3500 samples, Client 2 to Laying, Standing, Sitting, Walking with close to 2500 samples, and Client 3 to all six activities with close to 4500 samples. The visualization highlights how client-specific activity restrictions and uneven class counts produce strong non-IID conditions. This increases the difficulty of achieving a well-generalized global model in QFL.

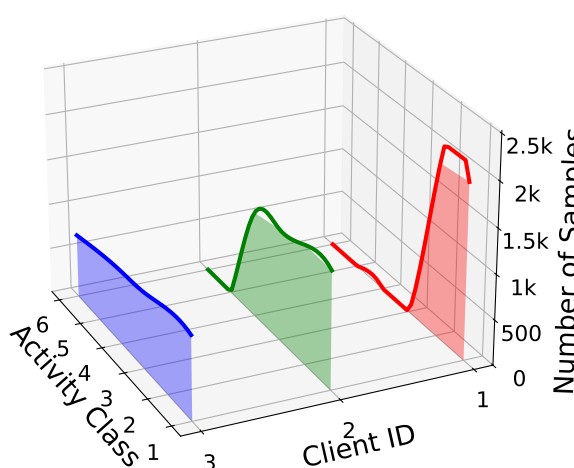

Figure 8: Heterogeneous distribution of HAR data among quantum clients. Each client has an imbalanced class distribution and a varying number of data instances compared to other clients.

### A.8 FUNDAMENTALS OF THE CLIENT-SIDE QUANTUM MODEL

#### A.8.1 DATA ENCODING

To be processed by the PQC, a classical data input $\mathbf{x}$ from the client's dataset must be encoded into the state of the circuit's $Q$ qubits. This is achieved using *amplitude encoding*. First, the $Q$ qubits are initialized in the ground state $|0\rangle$, resulting in the total initial state as

$$|\psi_{\text{initial}}\rangle = |0\rangle^{\otimes Q} \tag{32}$$

Let $\mathbf{x} = (x_0, x_1, \ldots, x_{2^Q-1})^\top$ be the real-valued feature vector, normalized such that $\sum_{i=0}^{2^Q-1} |x_i|^2 = 1$. The normalized vector is mapped directly to the amplitudes of the computational basis states given by

$$|\psi_{\text{encoded}}\rangle = \sum_{i=0}^{2^Q-1} x_i |i\rangle \tag{33}$$

The state equation 33 now contains the full classical feature vector in its amplitudes and serves as the input to the PQC.

#### A.8.2 PARAMETERIZED QUANTUM CIRCUIT (PQC)

The PQC processes the encoded state through a sequence of $L$ layers. The specific structure of the PQC for client $n$ is defined by its unique architecture matrix $\mathbf{A}_n \in \{0,1\}^{Q \times 3L}$. Each layer $l$ is composed of parameterized single-qubit rotation gates followed by a fixed block of entangling gates.

The transformation for a single layer $l$ is represented by a unitary operator $U_l$, which acts on the state from the previous layer $|\psi_{l-1}\rangle$ as

$$|\psi_l\rangle = U_l|\psi_{l-1}\rangle, \quad \text{where } |\psi_0\rangle = |\psi_{\text{encoded}}\rangle \tag{34}$$

The unitary $U_l$ is a composition of a rotation block $U_{\text{rot}}^{(l)}$ and an entanglement block $U_{\text{ent}}^{(l)}$, expressed as

$$U_l = U_{\text{ent}}^{(l)} U_{\text{rot}}^{(l)} \tag{35}$$

The rotation block $U_{\text{rot}}^{(l)}$ applies single-qubit gates to the $Q$ qubits. A gate $g \in \{R_x, R_y, R_z\}$ is applied to qubit $q$ at layer $l$ only if the corresponding entry $\mathbf{A}_{n(q,g,l)}$ in the architecture matrix is 1. Each applied gate is parameterized by a trainable angle $\Theta_n(q, g, l)$. The rotation gates are defined by the Pauli matrices $(X, Y, Z)$ as

$$R_g(\theta) = \exp\left(-i\frac{\theta}{2}g\right), \quad g \in \{X, Y, Z\} \tag{36}$$

The entanglement block $U_{\text{ent}}^{(l)}$ consists of fixed, non-parameterized two-qubit gates (e.g., CNOTs) that create correlations between the qubits.

The total unitary transformation performed by the client's PQC is given by

$$U_n(\theta_n) = \prod_{l=1}^{L} U_l \tag{37}$$

where $\theta_n$ represents the complete set of trainable parameters $\{\Theta_n(q, g, l)\}$. The final quantum state is then written as

$$|\psi_{\text{final}}\rangle = U_n(\theta_n)|\psi_{\text{encoded}}\rangle \tag{38}$$

### A.8.3 MEASUREMENT AND PREDICTION

To retrieve a classical result, the final state equation 38 is measured. This involves calculating the expectation value of the Pauli-$Z$ operator for each qubit $q$, expressed as

$$o_q = \langle\psi_{\text{final}}|Z_q|\psi_{\text{final}}\rangle \tag{39}$$

where $Z_q$ is the Pauli-$Z$ operator acting on qubit $q$. This process yields the classical output vector $\mathbf{o} = [o_1, o_2, \ldots, o_Q]^\top$.

### A.8.4 LOCAL TRAINING

Client $n$ trains its PQC by minimizing a local loss function $\mathcal{L}(\theta_n)$ that measures the discrepancy between the predictions and the true labels $y$. The gradient with respect to each parameter $\Theta_n(q, g, l)$ is computed using the parameter-shift rule:

$$\frac{\partial\mathcal{L}}{\partial\Theta_n(q, g, l)} \tag{40}$$

$$= \frac{1}{2}\left[\mathcal{L}\left(\theta_n + \frac{\pi}{2}\mathbf{e}_{(q,g,l)}\right) - \mathcal{L}\left(\theta_n - \frac{\pi}{2}\mathbf{e}_{(q,g,l)}\right)\right] \tag{41}$$

where $\mathbf{e}_{(q,g,l)}$ is a standard basis vector with one at the position corresponding to $\Theta_n(q, g, l)$ and zero elsewhere.

After computing the full gradient vector $\nabla\mathcal{L}(\theta_n^t)$, parameters are updated using the Adam optimizer. The update rule at iteration $t$ is:

$$\theta_n^{t+1} = \theta_n^t - \eta\nabla\mathcal{L}(\theta_n^t) \tag{42}$$

where $\eta$ is the learning rate. Local training proceeds for $E$ epochs before sending updated parameters to the central aggregator.

### A.8.5 QUANTUM NOISE

In BO-QFL, quantum noise arises primarily from two sources: (i) stochastic shot noise due to finite measurement sampling, and (ii) device-level depolarizing noise accumulated across clients. Both contribute additional error terms in the convergence analysis.

**Shot Noise:** Each client $n \in \mathcal{N}$ performs Pauli-$Z$ measurements with $H$ shots per expectation value. The variance of the unbiased estimator $\hat{Z}$ follows

$$\text{Var}[\hat{Z}] = \frac{1 - \langle Z\rangle^2}{H} \leq \frac{1}{H}.$$

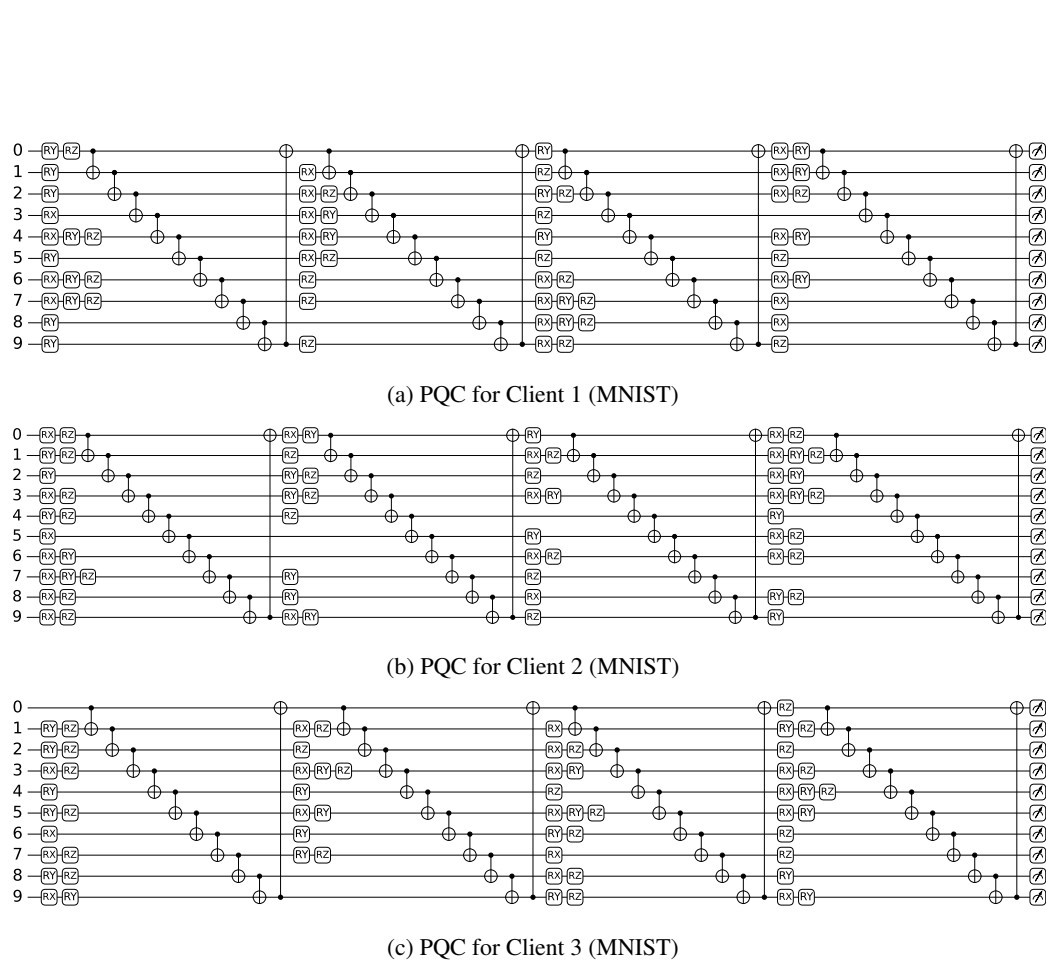

(a) PQC for Client 1 (MNIST)

(b) PQC for Client 2 (MNIST)

(c) PQC for Client 3 (MNIST)

Figure 9: BO-optimized parameterized quantum circuit of Client 1 for MNIST dataset, where 4 layers with different gate orientations are seen across 10 qubit wires. This figure contains only the PQC section of the QNN, excluding the encoder part.

Thus, increasing $H$ reduces the gradient variance linearly. In the convergence bound, this appears as

$$\frac{1}{N} \sum_{n \in \mathcal{N}} \frac{\nu N_z D \operatorname{Tr}(Z^2)}{2H},$$

where $D$ denotes the observable dimension, $N_z$ is the number of Pauli-$Z$ operators, and $\nu$ quantifies the variance constant.

**Depolarizing Noise:** Depolarizing noise Dür et al. (2005) is modeled by a quantum channel acting on a single-qubit state $\rho$ as

$$\mathcal{E}_{\text{dep}}(\rho) = (1-p)\rho + \frac{p}{2}I,$$

where $p \in [0.03, 0.05]$ is the depolarizing probability and $I$ is the identity operator. For $Q$-qubit PQCs, this channel extends as

$$\mathcal{E}_{\text{dep}}^{\otimes Q}(\rho) = (1-p)^Q \rho + \left(1 - (1-p)^Q\right)\frac{I}{2^Q}.$$

Under repeated circuit executions, the expectation value of an observable $O$ is biased toward the maximally mixed state:

$$\mathbb{E}[\langle O \rangle_{\text{noisy}}] = (1-p)^d \langle O \rangle_{\text{ideal}},$$

where $d$ is the circuit depth. For $N$ participating clients, this multiplicative attenuation accumulates across updates, and the aggregated gradient is effectively scaled by $(1-p)^{dN}$:

$$\nabla f(\bar{\boldsymbol{\theta}}_k^t)_{\text{noisy}} \approx (1-p)^{dN} \nabla f(\bar{\boldsymbol{\theta}}_k^t)_{\text{ideal}}.$$

In the convergence bound, this manifests as the fourth term, capturing the accumulated bias introduced by depolarizing noise.

**Interaction with BO and Aggregation:** Shot noise perturbs BO evaluations $f_n(\mathbf{A}_n)$, while depolarizing noise inflates the divergence term $\Psi_k^t$ in heterogeneous aggregation. Together, these effects validate Remark 2: although increasing $N$ and $H$ improves variance reduction and gradient averaging, excessive scaling amplifies accumulated depolarizing bias, resulting in diminishing or even negative returns in global performance.

## A.9    MODEL HETEROGENEITY

As an example, Figure 9 shows the BO-optimized PQCs for the three MNIST clients. These circuits were designed specifically to match the client-specific data distributions described earlier. Each model operates on 10 qubits, has four layers, and uses a fixed ring entanglement pattern in every layer. Although the overall structure is consistent, the placement and type of single-qubit rotation gates ($R_X$, $R_Y$, $R_Z$) differ between clients. These differences are the result of the BO search selecting gate arrangements that maximize performance for each client's data.

Figure 10 shows the BO-optimized PQCs for the three HAR clients, also tuned to the client-specific data distributions discussed earlier. Like the MNIST models, each PQC has 10 qubits, four layers, and a ring entanglement structure. The number and placement of rotation gates vary between clients, with some qubits having multiple rotations and others having fewer or none. These variations reflect BO's adaptation of each model to achieve the highest possible accuracy on its assigned HAR data.

## A.10    EXTENDED SIMULATION RESULTS

We study the client-level behavior during the BO-QFL process in this section, considering the framework with 3 quantum clients.

Local BO optimization was shown to improve clients' quantum neural networks' test accuracies substantially. Figure 11 demonstrates the progression of the optimal circuit design for each MNIST client over BO rounds, with large jumps representing BO rounds that yielded new optimal circuits, and flat periods representing rounds which did not yield new optimal circuits, but those rounds still played an important role in updating the surrogate model's (GP) representation of the relationship between circuit architecture and accuracy, guiding the search process. Client 1 steadily improved its

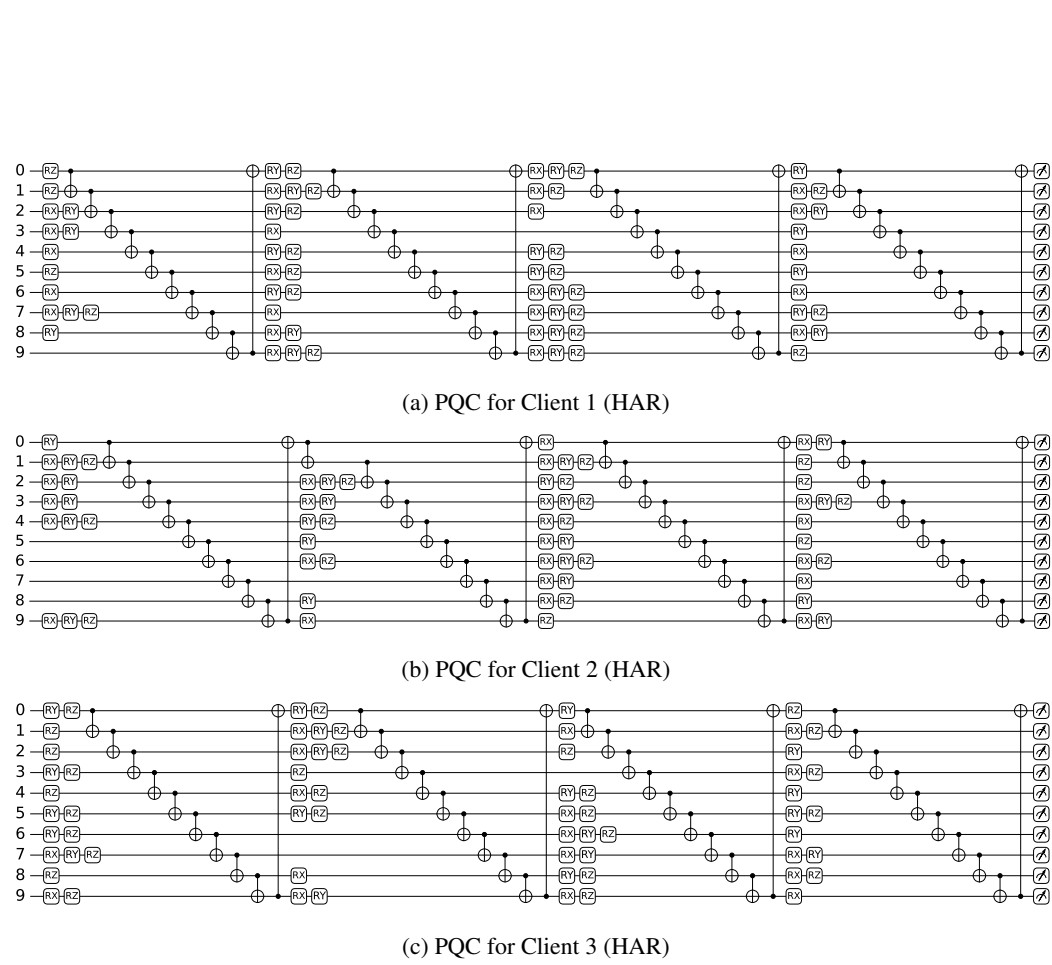

(a) PQC for Client 1 (HAR)

(b) PQC for Client 2 (HAR)

(c) PQC for Client 3 (HAR)

Figure 10: BO-optimized parameterized quantum circuit of Client 1 for HAR dataset, where 4 layers with different gate orientations are seen across 10 qubit wires. This figure contains only the PQC section of the QNN, excluding the encoder part.

optimal circuit from around 94.6% accuracy until converging at 100%, terminating BO earlier than 50 rounds. Client 1 realized a little over 5% test accuracy gain. Client 2 was the least optimized client across experiments, only managing to make an improvement once across 50 BO rounds, although he still found a decent improvement of about 2% test accuracy after less than 20 BO rounds. Client 3 made two steep jumps in test accuracy, improving up to 8% from the initial rounds, and finding an optimal accuracy after about 30 BO rounds, demonstrating the effectiveness of deep BO optimization as the largest improvement was realized late into the process.

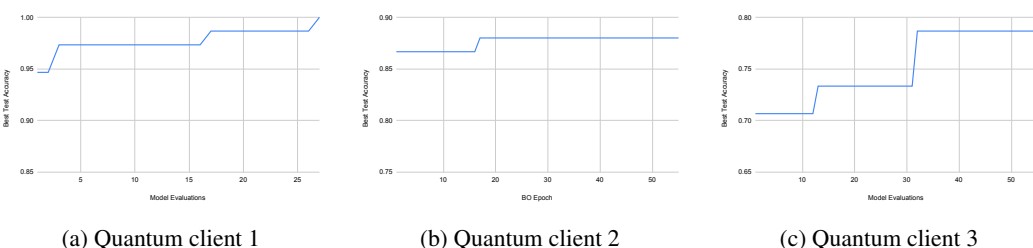

| (a) Quantum client 1 | (b) Quantum client 2 | (c) Quantum client 3 |

Figure 11: Accuracy progression and architecture updates over BO rounds for each client on the MNIST dataset. Each accuracy jump represents a new architecture that improved validation accuracy, while flat segments indicate rounds where no better configuration was found, highlighting the selective and adaptive behavior of the BO process in heterogeneous settings.

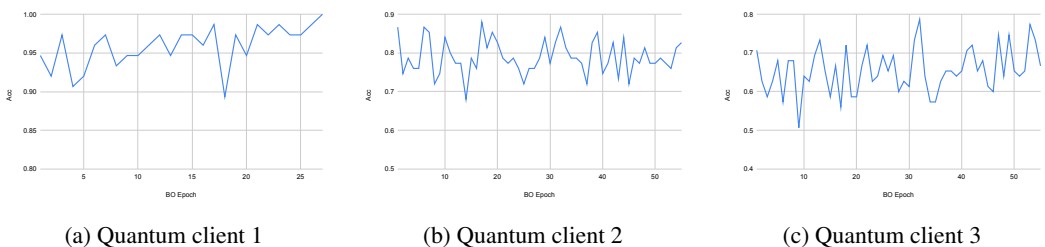

| (a) Quantum client 1 | (b) Quantum client 2 | (c) Quantum client 3 |

Figure 12: Accuracy search during BO rounds for each client on the MNIST dataset. Each accuracy jump represents a new architecture evaluated on validation accuracy.

Figure 12 visualizes the raw search process for each MNIST client, with each y-axis value being the accuracy of a circuit. As BO rounds progress, accuracy jumps all over as BO searches various circuits and updates the GP's belief of the search space. Large spikes represent BO rounds where a very good architecture was found and added to the dataset of observed circuit-accuracy pairs. For each client, accuracy fluctuates quite a large amount, implying BO effectively explores many areas of the search space and also implying that accuracy of the model on the clients test set is heavily dependent on the circuit architecture.

Figure 13 demonstrates the progression of the optimal circuit design for each HAR client, analogous to Figure 7 for MNIST. Client 1 made steep improvements in early rounds and then remained stagnant for most of the process, but found a substantial increase near the end, again validating the need for deep BO rounds. Client 1 eventually realized an overall gain of about 32% test accuracy. Client 2 similarly started out with steep improvements, but differs from Client 1 in the sense that further BO rounds did not help. Client 2 ends with an improvement of about 5% in test accuracy. Client 3 improves very sharply immediately and then improves steadily up until about 30 rounds, where it finds an optimum, overall realizing about a 10% test accuracy gain.

Figure 14 visualizes the raw BO search process for each HAR client, analogous to Figure 8 for MNIST. Fluctuations are not as large as in MNIST, implying either the search space was less effectively explored or the accuracy itself is less dependent on the circuit architecture. Nevertheless, quality optimization results were realized across clients, even if the process seemed to be slightly more stable.

Figures 15 and 16 show the loss comparisons between optimal architecture and standard architecture ($R_Y$) for all clients for MNIST and HAR, respectively. For MNIST (Figure 15), Client 1's loss

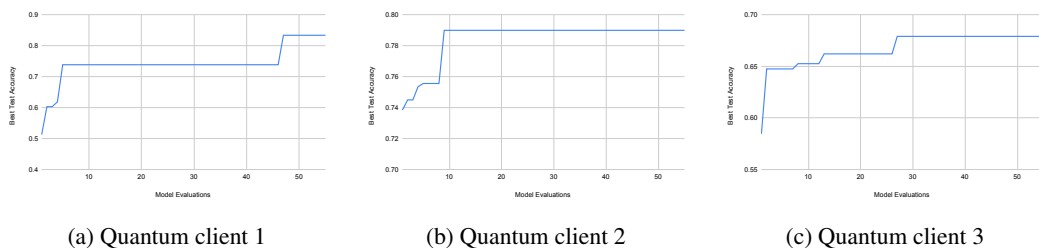

(a) Quantum client 1      (b) Quantum client 2      (c) Quantum client 3

Figure 13: Accuracy progression and architecture updates over BO rounds for each client on the HAR dataset. Each accuracy jump represents a new architecture that improved validation accuracy, while flat segments indicate rounds where no better configuration was found, highlighting the selective and adaptive behavior of the BO process in heterogeneous settings.

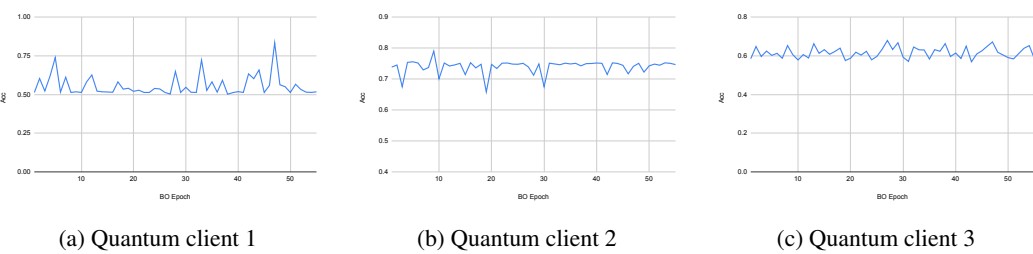

(a) Quantum client 1      (b) Quantum client 2      (c) Quantum client 3

Figure 14: Accuracy search during BO rounds for each client on the HAR dataset. Each accuracy jump represents a new architecture evaluated on validation accuracy.

curves are about the same, Client 2 the optimal architecture converge to a slightly lower loss, and Client 3's there is an even greater difference in the converged losses. For HAR (Figure 16), Client 1 shows an improvement in minimizing loss with the optimized circuit, Client 2 shows a slight improvement, and Client 3 shows a decent improvement.

## A.11 DETAILED ALGORITHMS

Algorithm 2 outlines the working procedure for the proposed BO-QFL framework. Each client runs a local architecture search using Algorithm 3 (lines 3–4), after which the server unifies architectures and initializes weights (lines 6-7). Across rounds, clients train locally and send updates (lines 8–10), while the server aggregates parameters using Algorithm 4 and redistributes them (lines 12-13). This repeats until convergence or the round limit is reached (lines 14-17).

Algorithm 3 summarizes the BO process for QNN architecture optimization. It first samples and evaluates initial architectures to build the dataset (lines 3–4). In each round, it fits the GP surrogate model (lines 5-7), computes LogEI (line 8), selects and evaluates a new candidate (lines 9-10),

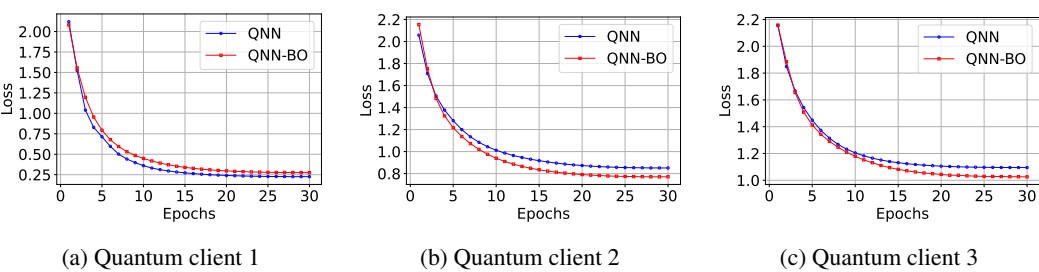

(a) Quantum client 1      (b) Quantum client 2      (c) Quantum client 3

Figure 15: Training loss comparison between BO-optimized (QNN-BO) and traditional QNN architectures across individual clients on the MNIST dataset.

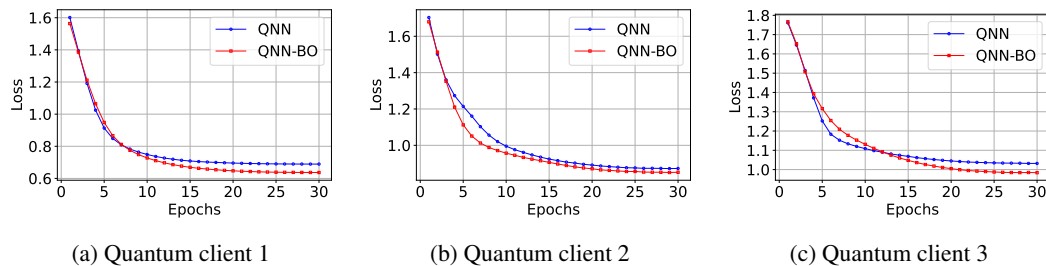

(a) Quantum client 1                    (b) Quantum client 2                    (c) Quantum client 3

Figure 16: Training loss comparison between BO-optimized (QNN-BO) and traditional QNN architectures across individual clients on the HAR dataset.

---

**Algorithm 2** BO-QFL System

---

1: **Input:** $N$ clients with local datasets $\{D_n\}$, PQC search space $\mathcal{S}$, communication rounds $K$
2: **Output:** Global model $\theta_G$ and architecture $\mathbf{A}_G$
3: **for** each client $n \in \mathcal{N}$ **in parallel do**
4:       Local architecture search via Algorithm 3 $\rightarrow \mathbf{A}_n^*$
5: **end for**
6: Server constructs global architecture $\mathbf{A}_G$ via union rule.
7: Initialize global PQC weights to all clients as in equation 2
8: **for** each round $k = 1$ to $K$ **do**
9:       **for** each client $n \in \mathcal{N}$ **in parallel do**
10:           Train local PQC for $E$ epochs & send updates.
11:       **end for**
12:       Server aggregates parameters using Algorithm 4 $\rightarrow \theta_G$
13:       Each client updates its local model with $\theta_G$.
14:       **if** converged **then**
15:           **break**
16:       **end if**
17: **end for**

---

**Algorithm 3** Bayesian Optimization for QNN Architecture Search

---

1: **Input:** $n_0$ (initial samples), $E$ (max evaluations), $Q \times 3L$ (architecture size), client data
2: **Output:** Optimized architecture $\mathbf{x}^* = \arg\max_{\mathbf{x}_i \in \mathcal{D}_e} f(\mathbf{x}_i)$, $y^{\max} = \max_i f(\mathbf{x}_i)$
3: Sample $n_0$ initial architecture vectors $\mathbf{x}_i$ from $[0, 1]^{Q \times 3L}$ via Sobol sequence
4: Round each $\mathbf{x}_i$ to binary $\{0, 1\}^{Q \times 3L}$
5: Train QNN for each $\mathbf{x}_i$, record accuracy $f(\mathbf{x}_i)$, set $\mathcal{D}_0 = \{(\mathbf{x}_i, f(\mathbf{x}_i))\}$
6: **for** $e = n_0 + 1$ to $E$ **do**
7:       Fit Gaussian Process surrogate model by maximizing marginal II.
8:       Calculate Log Expected Improvement.
9:       Select next candidate $\mathbf{x}_e = \arg\max_{\mathbf{x}} \text{LogEI}(\mathbf{x})$
10:       Round $\mathbf{x}_e$ to binary, train QNN, record $f(\mathbf{x}_e)$
11:       Update data: $\mathcal{D}_t = \mathcal{D}_{e-1} \cup \{(\mathbf{x}_e, f(\mathbf{x}_e))\}$
12:       **if** stopping criterion met (converged or $e = E$) **then**
13:           **break**
14:       **end if**
15: **end for**

---

and updates the dataset (line 11). This loop continues until convergence (line 12) or reaching the evaluation limit.

Algorithm 4 details the strategy for heterogeneous model aggregation. It begins by initializing each client's parameter set and an empty tensor for the new global model (lines 2-3). For every parameter position in the global set, it identifies which clients trained that parameter (line 5), then averages their values (line 6) to update that parameter in the global model tensor.

---

**Algorithm 4** Heterogeneous Model Aggregation

---

1: **Input:** $\mathcal{N}$, $\{\mathbf{A}_n^*\}_{n \in \mathcal{N}}$, $\{\Theta_n^{(k+1)}\}_{n \in \mathcal{N}}$.
2: **Output:** Updated global model parameters $\theta_G^{(k+1)}$.
3: **Initialization:**
    Client parameter sets $\mathcal{W}_n^*$ using equation **??**, global parameter set
    $\mathcal{W}_G$, and empty tensor for the new global model, $\theta_G^{(k+1)}$.
4: **for** each parameter position $p \in \mathcal{W}_G$ **do**
5:     Identify the specific set of clients that trained this
      parameter: $\mathcal{N}_p \leftarrow \{n \in \mathcal{N} \mid p \in \mathcal{W}_n^*\}$.
6:     Calculate the new global parameter via Equation equation 3 and
      unify them $\rightarrow$ updated $\theta_G^{(k+1)}$.
7: **end for**

---

