# OpenReview forum: "Heterogeneous Quantum Federated Learning via Adaptive Circuit Search and Model Aggregation"
_ICLR.cc/2026/Conference — ICLR 2026 Conference Withdrawn Submission_

### Official Review · Reviewer_xALw · 2025-10-24

**Soundness:** 2
**Presentation:** 2
**Contribution:** 2
**Rating:** 2
**Confidence:** 4

**Summary:**

The paper introduces BO-QFL, a framework for heterogeneous quantum federated learning (QFL) that addresses two limitations in existing QFL systems: the use of identical quantum circuit architectures across clients and the absence of an aggregation mechanism for structurally distinct models. The paper proposes (i) a Bayesian Optimization (BO)-based search for client-specific quantum circuit architectures, and (ii) a heterogeneous model aggregation rule using element-wise logical unions to unify diverse parameterized quantum circuits. The paper includes a detailed convergence analysis, extensive simulation results on MNIST, Fashion-MNIST, and HAR datasets, and comparisons with both reinforcement learning (RL)-based architecture search and baseline QFL approaches.

**Strengths:**

1.	The paper introduces an original combination of Bayesian optimization with federated quantum learning.
2.	The paper provides a formal convergence analysis, albeit under strong simplifying assumptions.
3.	The method is evaluated on multiple datasets and the results report consistent performance improvements over a baseline QFL model.

**Weaknesses:**

1.	In the convergence analysis in Section 5, the proof framework is nearly identical to that of Ajalloeian & Stich (2020) and FedAvg variants, with only superficial adaptation to “quantum heterogeneity.” No empirical verification of the claimed convergence rate is presented, making the theoretical section largely disconnected from the proposed BO-QFL mechanism.
2.	The algorithmic pipeline in Figure 1 and Algorithm 1 is conceptually unclear. The server aggregation rule (Eq. 3) is presented without justification of its stability or gradient equivalence to a consistent loss function, raising doubts about whether the aggregation preserves differentiability or introduces bias in parameter updates.
3.	The experiments rely entirely on classical simulations using TorchQuantum and PyTorch on a GPU, without validating the method on an actual quantum processor. The reported “quantum noise” is synthetically modeled, making it difficult to claim robustness in real-world settings.
4.	In Table 1, the numerical results are modest and occasionally inconsistent. For instance, in HAR dataset results, BO-QFL’s accuracy decreases with more clients even in ideal settings, which contradicts the stated theoretical remark (Remark 2). No discussion is provided to explain this anomaly.
5.	The assumed client model heterogeneity (up to 12 clients, 10 qubits each) is small relative to real-world FL settings. The communication and circuit reconstruction costs are not quantified, so scalability claims are speculative.
6.	The two stated contributions (adaptive BO circuit search and heterogeneous aggregation) are intertwined in the implementation, making it unclear which component drives the performance gains. An ablation study is required to identify the gains derived from each contribution.

**Questions:**

1.	How does the proposed aggregation (Eq. 3) preserve gradient consistency across heterogeneous circuits? Has this been validated empirically?
2.	Could the paper quantify the computational cost of Bayesian optimization compared to RL in terms of wall-clock time, not only the number of evaluations?
3.	How would BO-QFL behave under partial client participation or asynchronous updates?

---

### Official Review · Reviewer_ruBB · 2025-10-31

**Soundness:** 2
**Presentation:** 2
**Contribution:** 2
**Rating:** 2
**Confidence:** 5

**Summary:**

In this work, the authors explore how quantum federated learning (QFL) can be made more flexible and effective when quantum devices differ in both hardware capacity and data distribution. The authors note that existing QFL systems assume that all quantum clients use the same quantum circuit design, which is unrealistic when devices have different numbers of qubits or data types. To address this issue, they propose a framework called BO-QFL that automatically enables each client to identify its best-performing quantum circuit architecture using Bayesian optimization. These locally optimized circuits are then combined at a central server through a new aggregation method that merges all clients’ circuit structures using a logical “union” rule and averages parameters only where applicable.

**Strengths:**

1. The paper focuses on a real challenge in quantum federated learning, so that different quantum devices (clients) have different hardware limits and data types, so one shared circuit design doesn’t fit all.

2. The authors provide a theoretical analysis to show that the algorithm is stable and can converge under reasonable conditions, giving the method more credibility.

3. The proposed BO-QFL framework allows each client to automatically find its own best quantum circuit using Bayesian optimization instead of forcing everyone to use the same one.

**Weaknesses:**

1. The proposed BO-QFL mainly combines two known ideas: Bayesian optimization for quantum circuit search and masked averaging for model aggregation. The integration is logical but not fundamentally new compared to existing quantum federated learning (QFL) or quantum architecture search works.

2. All experiments are done on classical simulators rather than real quantum hardware, so the framework’s ability to handle noise, qubit limitations, and circuit execution costs in practice is not truly verified.

3. The “union” of different client circuits may produce a much larger and deeper global circuit, which could become inefficient or even infeasible to implement on real devices. The paper provides no analysis or results on this issue.

4. The convergence proof relies on strong assumptions (e.g., smoothness, Polyak-Łojasiewicz condition, bounded bias of Bayesian optimization) that make the result theoretically correct but practically weak. The analysis does not cover how heterogeneous circuits change these bounds.

5. The benchmarks (MNIST, Fashion-MNIST, HAR) are small and classical datasets, so the results do not convincingly demonstrate QFL advantages on large-scale or quantum-native problems.

6. The paper does not report the runtime, number of evaluations, or communication overhead of Bayesian optimization across clients, leaving uncertainty about real-world feasibility.

**Questions:**

1. How large does the global “union” circuit become? In particular, when each client has its own circuit structure, the logical “union” might make the global circuit much larger. Could the authors provide quantitative results on how the circuit depth and gate count scale with the number of clients?

2. How is hardware heterogeneity actually simulated? The paper claims to model clients with different qubit resources, but all tests seem to use classical simulators. How exactly do you simulate differences in qubit counts or noise levels across clients?

3. Why is Bayesian optimization preferred over reinforcement learning or gradient-based circuit search? What advantages does BO offer in this federated quantum context?

4. What happens if two clients find very different optimal architectures? How does the aggregation rule handle extreme cases where there is little overlap between client circuits? Does performance degrade as architectural diversity grows?

5. How are the local objectives for Bayesian optimization defined? Is the optimization guided by validation accuracy, loss, or fidelity metrics? And how many evaluations are needed for each local BO step?

6. How much communication and computation overhead does BO-QFL introduce? Since each client runs its own optimization and sends its parameters back to the server, could the authors provide a timing or cost analysis compared with the standard FedAvg-QFL?

---

### Official Review · Reviewer_eeHq · 2025-11-01

**Soundness:** 3
**Presentation:** 3
**Contribution:** 2
**Rating:** 2
**Confidence:** 4

**Summary:**

The goal of this paper is to develop a quantum federated learning algorithm that can  handle heterogeneity. The authors address this problem by leveraging techniques from Bayesian optimization, then they provide analytical and experimental results.

**Strengths:**

+ The idea of using Bayesian optimization to find optimal quantum circuits is meaningful.
+ Heterogeneity is a problem less explored in the context of QFL and thus the work is timely.
+ The theoretical results are rigorous.

**Weaknesses:**

- Despite their rigor, the theoretical results provide very limited new insights.
- The aggregation seems to follow an incremental version of FedAvg and, thus, it is not clear if it is the most effective way to capture heterogeneous systems.
- It is not clear why one needs a federated learning in this space of quantum networks.
- The experimental results are very limited to basic MNIST datasets. The baselines are also very rudimentary.

**Questions:**

- What is the benefit from the insights gained out of Theorem 1? It seems these are known notion in classical FL. What is unique to quantum here?
- Can you modify your algorithm to introduce a more effective aggregation rule?
- Can your approach properly handle non-IIDness?
- Can you expand the experiments on more advanced datasets like CIFAR?
- What is the complexity of the BO piece?
- What are the quantum hardware requirements for your approach to actually be deployed?
- What is the communication overhead and would you require entanglement?

---

### Note · Authors · 2025-11-14

I have read and agree with the venue's withdrawal policy on behalf of myself and my co-authors.